# GENERALIZED ENERGY BASED MODELS

**Michael Arbel**[*]**, Liang Zhou & Arthur Gretton**
Gatsby Computational Neuroscience Unit, University College London

## ABSTRACT

We introduce the Generalized Energy Based Model (GEBM) for generative modelling. These models combine two trained components: a base distribution (generally an implicit model), which can learn the support of data with low intrinsic dimension in a high dimensional space; and an energy function, to refine the probability mass on the learned support. Both the energy function and base jointly constitute the final model, unlike GANs, which retain only the base distribution (the "generator"). GEBMs are trained by alternating between learning the energy and the base. We show that both training stages are well-defined: the energy is learned by maximising a generalized likelihood, and the resulting energy-based loss provides informative gradients for learning the base. Samples from the posterior on the latent space of the trained model can be obtained via MCMC, thus finding regions in this space that produce better quality samples. Empirically, the GEBM samples on image-generation tasks are of much better quality than those from the learned generator alone, indicating that all else being equal, the GEBM will outperform a GAN of the same complexity. When using normalizing flows as base measures, GEBMs succeed on density modelling tasks, returning comparable performance to direct maximum likelihood of the same networks.

## 1 INTRODUCTION

Energy-based models (EBMs) have a long history in physics, statistics and machine learning (LeCun et al., 2006). They belong to the class of *explicit* models, and can be described by a family of energies $E$ which define probability distributions with density proportional to $\exp(-E)$. Those models are often known up to a normalizing constant $Z(E)$, also called the *partition function*. The learning task consists of finding an optimal function that best describes a given system or target distribution $\mathbb{P}$. This can be achieved using maximum likelihood estimation (MLE), however the intractability of the normalizing partition function makes this learning task challenging. Thus, various methods have been proposed to address this (Hinton, 2002; Hyvärinen, 2005; Gutmann and Hyvärinen, 2012; Dai et al., 2019a;b). All these methods estimate EBMs that are supported over the whole space. In many applications, however, $\mathbb{P}$ is believed to be supported on an unknown lower dimensional manifold. This happens in particular when there are strong dependencies between variables in the data (Thiry et al., 2021), and suggests incorporating a low-dimensionality hypothesis in the model.

Generative Adversarial Networks (GANs) (Goodfellow et al., 2014) are a particular way to enforce low dimensional structure in a model. They rely on an *implicit* model, the generator, to produce samples supported on a low-dimensional manifold by mapping a pre-defined latent noise to the sample space using a trained function. GANs have been very successful in generating high-quality samples on various tasks, especially for unsupervised image generation (Brock et al., 2018). The generator is trained *adversarially* against a discriminator network whose goal is to distinguish samples produced by the generator from the target data. This has inspired further research to extend the training procedure to more general losses (Nowozin et al., 2016; Arjovsky et al., 2017; Li et al., 2017; Bińkowski et al., 2018; Arbel et al., 2018) and to improve its stability (Miyato et al., 2018; Gulrajani et al., 2017; Nagarajan and Kolter, 2017; Kodali et al., 2017). While the generator of a GAN has effectively a low-dimensional support, it remains challenging to refine the distribution of mass on that support using pre-defined latent noise. For instance, as shown by Cornish et al. (2020) for normalizing flows, when the latent distribution is unimodal and the target distribution possesses multiple disconnected low-dimensional components, the generator, as a continuous map, compensates for this mismatch using steeper slopes. In practice, this implies the need for more complicated generators.

---

[*]Correspondence: `michael.n.arbel@gmail.com`.

In the present work, we propose a new class of models, called *Generalized Energy Based Models* (GEBMs), which can represent distributions supported on low-dimensional manifolds, while offering more flexibility in refining the mass on those manifolds. GEBMs combine the strength of both *implicit* and *explicit* models in two separate components: a base distribution (often chosen to be an implicit model) which learns the low-dimensional support of the data, and an energy function that can refine the probability mass on that learned support. We propose to train the GEBM by alternating between learning the energy and the base, analogous to $f$-GAN training (Goodfellow et al., 2014; Nowozin et al., 2016). The energy is learned by maximizing a generalized notion of likelihood which we relate to the *Donsker-Varadhan* lower-bound (Donsker and Varadhan, 1975) and *Fenchel duality*, as in (Nguyen et al., 2010; Nowozin et al., 2016). Although the partition function is intractable in general, we propose a method to learn it in an amortized fashion without introducing additional surrogate models, as done in variational inference (Kingma and Welling, 2014; Rezende et al., 2014) or by Dai et al. (2019a;b). The resulting maximum likelihood estimate, the *KL Approximate Lower-bound Estimate (*KALE*)*, is then used as a loss for training the base. When the class of energies is rich and smooth enough, we show that KALE leads to a meaningful criterion for measuring weak convergence of probabilities. Following recent work by Chu et al. (2020); Sanjabi et al. (2018), we show that KALE possesses well defined gradients w.r.t. the parameters of the base, ensuring well-behaved training. We also provide convergence rates for the empirical estimator of KALE when the variational family is sufficiently well behaved, which may be of independent interest.

The main advantage of GEBMs becomes clear when sampling from these models: the posterior over the latents of the base distribution incorporates the learned energy, putting greater mass on regions in this latent space that lead to better quality samples. Sampling from the GEBM can thus be achieved by first sampling from the posterior distribution of the latents via MCMC in the low-dimensional latent space, then mapping those latents to the input space using the implicit map of the base. This is in contrast to standard GANs, where the latents of the base have a fixed distribution. We focus on a class of samplers that exploit gradient information, and show that these samplers enjoy fast convergence properties by leveraging the recent work of Eberle et al. (2017). While there has been recent interest in using the discriminator to improve the quality of the generator during sampling (Azadi et al., 2019; Turner et al., 2019; Neklyudov et al., 2019; Grover et al., 2019; Tanaka, 2019; Wu et al., 2019b), our approach emerges naturally from the model we consider.

We begin in Section 2 by introducing the GEBM model. In Section 3, we describe the learning procedure using KALE, then derive a method for sampling from the learned model in Section 4. In Section 5 we discuss related work. Finally, experimental results are presented in Section 6 with code available at `https://github.com/MichaelArbel/GeneralizedEBM`.

## 2 GENERALIZED ENERGY-BASED MODELS

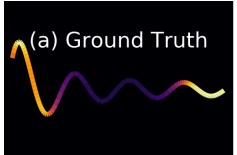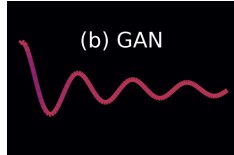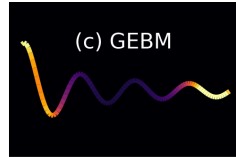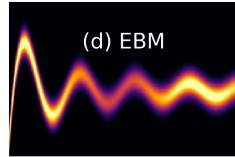

Figure 1: Data generating distribution supported on a line and with higher density at the extremities. Models are learned using either a GAN, GEBM, or EBM. More details are provided in Appendix G.3.

In this section, we introduce generalized energy based models (GEBM), that combine the strengths of both energy-based models and implicit generative models, and admit the first of these as a special case. An **energy-based model** (EBM) is defined by a set $\mathcal{E}$ of real valued functions called *energies*, where each $E \in \mathcal{E}$ specifies a probability density over the data space $\mathcal{X} \subset \mathbb{R}^d$ up to a normalizing constant,

$$\mathbb{Q}(\mathrm{d}x) = \exp(-E(x) - A)\mathrm{d}x, \qquad A = \log\left(\int \exp(-E(x))\mathrm{d}x\right). \qquad (1)$$

While EBMs have been shown recently to be powerful models for representing complex high dimensional data distributions, they still unavoidably lead to a blurred model whenever data are concentrated on a lower-dimensional manifold. This is the case in Figure 1(a), where the ground truth distribution is

supported on a 1-D line and embedded in a 2-D space. The EBM in Figure 1(d) learns to give higher density to a halo surrounding the data, and thus provides a blurred representation. That is a consequence of EBM having a density defined over the whole space, and can result in blurred samples for image models.

An **implicit generative model** (IGM) is a family of probability distributions $\mathbb{G}_\theta$ parametrized by a learnable *generator* function $G : \mathcal{Z} \mapsto \mathcal{X}$ that maps latent samples $z$ from a fixed latent distribution $\eta$ to the data space $\mathcal{X}$. The latent distribution $\eta$ is required to have a density over the latent space $\mathcal{Z}$ and is often easy to sample from. Thus, Sampling from $\mathbb{G}$ is simply achieved by first sampling $z$ from $\eta$ then applying $G$,

$$x \sim \mathbb{G} \iff x = G(z), \quad z \sim \eta. \tag{2}$$

GANs are popular instances of these models, and are trained *adversarially* (Goodfellow et al., 2014). When the latent space $\mathcal{Z}$ has a smaller dimension than the input space $\mathcal{X}$, the IGM will be supported on a lower dimensional manifold of $\mathcal{X}$, and thus will not possess a Lebesgue density on $\mathcal{X}$ (Bottou et al., 2017). IGMs are therefore good candidates for modelling low dimensional distributions. While GANs can accurately learn the low-dimensional support of the data, they can have limited power for representing the distribution of mass on the support. This is illustrated in Figure 1(b).

A **generalized energy-based model** (GEBM) $\mathbb{Q}$ is defined by a combination of a *base* $\mathbb{G}$ and an *energy* $E$ defined over a subset $\mathcal{X}$ of $\mathbb{R}^d$. The **base** component can typically be chosen to be an IGM as in (2). The **generalized energy** component can refine the mass on the support defined by the *base*. It belongs to a class $\mathcal{E}$ of real valued functions defined on the input space $\mathcal{X}$, and represents the negative log-density of a sample from the GEBM with respect to the base $\mathbb{G}$,

$$\mathbb{Q}(\mathrm{d}x) = \exp(-E(x) - A_{\mathbb{G},E})\mathbb{G}(\mathrm{d}x), \qquad A_{\mathbb{G},E} = \log\left(\int \exp(-E(x))\mathbb{G}(\mathrm{d}x)\right), \tag{3}$$

where $A_{\mathbb{G},E}$ is the logarithm of the normalizing constant of the model w.r.t. $\mathbb{G}$. Thus, a GEBM $\mathbb{Q}$ re-weights samples from the base according to the un-normalized importance weights $\exp(-E(x))$. Using the latent structure of the base $\mathbb{G}$, this importance weight can be pulled-back to the latent space to define a *posterior latent* distribution $\nu$,

$$\nu(z) := \eta(z)\exp(-E(G(z)) - A_{\mathbb{G},E}). \tag{4}$$

Hence, the *posterior latent* $\nu$ can be used instead of the latent noise $\eta$ for sampling from $\mathbb{Q}$, as summarized by Proposition 1:

**Proposition 1.** *Sampling from $\mathbb{Q}$ requires sampling a latent $z$ from $\nu$ (4) then applying the map $G$,*

$$x \sim \mathbb{Q} \iff x = G(z), \quad z \sim \nu. \tag{5}$$

In order to hold, Proposition 1 does not need the generator $G$ to be invertible. We provide a proof in Appendix C.1 which relies on a characterization of probability distribution using generalized moments. We will see later in Section 4 how equation (5) can be used to provide practical sampling algorithms from the GEBM. Next we discuss the advantages of GEBMs.

**Advantages of Generalized Energy Based Models.** The GEBM defined by (3) can be related to exponential tilting (re-weighting) (Siegmund, 1976; Xie et al., 2016) of the base $\mathbb{G}$. The important difference over classical EBMs is that the base $\mathbb{G}$ is allowed to change its support and shape in space. By learning the base $\mathbb{G}$, GEBMs can accurately learn the low-dimensional support of data, just like IGMs do. They also benefit from the flexibility of EBMs for representing densities using an energy $E$ to refine distribution of mass on the support defined by $\mathbb{G}$, as seen in Figure 1(c).

**Compared to EBMs**, that put mass on the whole space by construction (positive density), GEBMs have the additional flexibility to concentrate the probability mass on a low-dimensional support learned by the base $\mathbb{G}$, provided that the dimension of the latent space $\mathcal{Z}$ is smaller than the dimension of the ambient space $\mathcal{X}$: see Figure 1(c) vs Figure 1(d). In the particular case when the dimension of $\mathcal{Z}$ is equal to the ambient dimension and $G$ is invertible, the base $\mathbb{G}$ becomes supported over the whole space $\mathcal{X}$, and GEBM recover usual EBMs. The next proposition further shows that any EBM can be viewed as a particular cases of GEBMs, as proved in Appendix C.1.

**Proposition 2.** *Any EBM with energy $E$ (as in (1)) can be expressed as a GEBM with base $\mathbb{G}$ given as a normalizing flow with density $\exp(-r(x))$ and a generalized energy $\tilde{E}(x) = E(x) - r(x)$. In this particular case, the dimension of the latent is necessarily equal to the data dimension, i.e. $\dim(\mathcal{Z}) = \dim(\mathcal{X})$.*

**Compared to IGMs**, that rely on a fixed pre-determined latent noise distribution $\eta$, GEBMs offer the additional flexibility of learning a richer latent noise distribution. This is particularly useful when the data is multimodal. In IGMs, such a GANs, the latent noise $\eta$ is usually unimodal thus requiring a more sophisticated generator to distort a unimodal noise distribution into a distribution with multiple modes, as shown by Cornish et al. (2020). Instead, GEBMs allow to sample from a *posterior $\nu$* over the latent noise defined in (4). This posterior noise can be multimodal in latent space (by incorporating information from the energy) and thus can put more or less mass in specific regions of the manifold defined by the base $\mathbb{G}$. This allows GEBMs to capture multimodality in data, provided the support of the base is broad enough to subsume the data support Figure 1(c). The base can be simpler, compared to GANs, as it doesn't need to distort the input noise too much to produce multimodal samples (see Figure 8 in Appendix G.4). This additional flexibility comes at no additional training cost compared to GANs. Indeed, GANs still require another model during training, the discriminator network, but do not use it for sampling. Instead, GEBMs avoid this waist since the base and energy can be trained jointly, with no other additional model, and then both are used for sampling.

## 3    LEARNING GEBMS

In this section we describe a general procedure for learning GEBMs. We decompose the learning procedure into two steps: an *energy learning* step and a *base learning* step. The overall learning procedure alternates between these two steps, as done in GAN training (Goodfellow et al., 2014).

### 3.1    ENERGY LEARNING

When the base $\mathbb{G}$ is fixed, varying the energy $E$ leads to a family of models that all admit a density $\exp(-E - A_{\mathbb{G},E})$ w.r.t. $\mathbb{G}$. When the base $\mathbb{G}$ admits a density $\exp(-r)$ defined over the whole space, it is possible to learn the energy $E$ by maximizing the likelihood of the model $-\int(E+r)\mathrm{d}\mathbb{P} - A_{\mathbb{G},E}$. However, in general $\mathbb{G}$ is supported on a lower-dimensional manifold so that $r$ is ill-defined and the usual notion of likelihood cannot be used. Instead, we introduce a generalized notion of likelihood which does not require a well defined density $\exp(-r)$ for $\mathbb{G}$:

**Definition 1** (Generalized Likelihood). *The expected $\mathbb{G}$-log-likelihood under a target distribution $\mathbb{P}$ of a GEBM model $\mathbb{Q}$ with base $\mathbb{G}$ and energy $E$ is defined as*

$$\mathcal{L}_{\mathbb{P},\mathbb{G}}(E) := -\int E(x)d\mathbb{P}(x) - A_{\mathbb{G},E}. \tag{6}$$

To provide intuitions about the generalized likelihood in Definition 1, we start by discussing the particular case where $KL(\mathbb{P}||\mathbb{G}) < +\infty$. We then present the training method in the general case where $\mathbb{P}$ and $\mathbb{G}$ might not share the same support, i.e. $KL(\mathbb{P}||\mathbb{G}) = +\infty$.

**Special case of finite $KL(\mathbb{P}||\mathbb{G})$.** When the Kullback-Leibler divergence between $\mathbb{P}$ and $\mathbb{G}$ is well defined, (6) corresponds to the Donsker-Varadhan (DV) lower bound on the KL (Donsker and Varadhan, 1975), meaning that $\mathrm{KL}(\mathbb{P}||\mathbb{G}) \geq \mathcal{L}_{\mathbb{P},\mathbb{G}}(E)$ for all $E$. Moreover, the following proposition holds:

**Proposition 3.** *Assume that $KL(\mathbb{P}||\mathbb{G}) < +\infty$ and $0 \in \mathcal{E}$. If, in addition, $E^\star$ maximizes (6), then:*

$$KL(\mathbb{P}||\mathbb{Q}) \leq KL(\mathbb{P}||\mathbb{G}). \tag{7}$$

*In addition, we have that $KL(\mathbb{P}||\mathbb{Q}) = 0$ when $E^\star$ is the negative log-density ratio of $\mathbb{P}$ w.r.t. $\mathbb{G}$.*

We refer to Appendix C.1 for a proof. According to (7), the GEBM systematically improves over the IGM defined by $\mathbb{G}$, with no further improvement possible in the limit case when $\mathbb{G} = \mathbb{P}$. Hence as long as there is an error in mass on the common support of $\mathbb{P}$ and $\mathbb{G}$, the GEBM improves over the base $\mathbb{G}$.

**Estimating the likelihood in the General setting.** Definition 1 can be used to learn a maximum likelihood energy $E^\star$ by maximizing $\mathcal{L}_{\mathbb{P},\mathbb{G}}(E)$ w.r.t. $E$ even when the $KL(\mathbb{P}||\mathbb{G})$ is infinite and when $\mathbb{P}$ and $\mathbb{G}$ don't necessarily share the same support. Such an optimal solution is well defined whenever the set of energies is suitably constrained. This is the case if the energies are parametrized by a compact set $\Psi$ with $\psi \mapsto E_\psi$ continuous over $\Psi$. Estimating the likelihood is then achieved using i.i.d. samples $(X_n)_{1:N}, (Y_m)_{1:M}$ from $\mathbb{P}$ and $\mathbb{G}$ (Tsuboi et al., 2009; Sugiyama et al., 2012; Liu et al., 2017):

$$\hat{\mathcal{L}}_{\mathbb{P},\mathbb{G}}(E) = -\frac{1}{N}\sum_{n=1}^{N} E(X_n) - \log\left(\frac{1}{M}\sum_{m=1}^{M}\exp(-E(Y_m))\right). \tag{8}$$

In the context of mini-batch stochastic gradient methods, however, $M$ typically ranges from 10 to 1000, which can lead to a poor estimate for the log-partition function $A_{\mathbb{G},E}$. Moreover, (8) doesn't exploit estimates of $A_{\mathbb{G},E}$ from previous gradient iterations. Instead, we propose an estimator which introduces a variational parameter $A \in \mathbb{R}$ meant to estimate $A_{\mathbb{G},E}$ in an amortized fashion. The key idea is to exploit the convexity of the exponential which directly implies $-A_{\mathbb{G},E} \geq -A - \exp(-A + A_{\mathbb{G},E}) + 1$ for any $A \in \mathbb{R}$, with equality only when $A = A_{\mathbb{G},E}$. Therefore, (6) admits a lower-bound of the form

$$\mathcal{L}_{\mathbb{P},\mathbb{G}}(E) \geq -\int (E+A)\mathrm{d}\mathbb{P} - \int \exp(-(E+A))\mathrm{d}\mathbb{G} + 1 := \mathcal{F}_{\mathbb{P},\mathbb{G}}(E+A),$$

where we introduced the functional $\mathcal{F}_{\mathbb{P},\mathbb{G}}$ for concision. Maximizing $\mathcal{F}_{\mathbb{P},\mathbb{G}}(E+A)$ over $A$ recovers the likelihood $\mathcal{L}_{\mathbb{P},\mathbb{G}}(E)$. Moreover, jointly maximizing over $E$ and $A$ yields the maximum likelihood energy $E^\star$ and its corresponding log-partition function $A^\star = A_{\mathbb{G},E^\star}$. This optimization is well-suited for stochastic gradient methods using the following estimator Kanamori et al. (2011):

$$\hat{\mathcal{F}}_{\mathbb{P},\mathbb{G}}(E+A) = -\frac{1}{N}\sum_{n=1}^{N}(E(X_n)+A) - \frac{1}{M}\sum_{m=1}^{M}\exp(-(E(Y_m)+A)) + 1. \tag{9}$$

### 3.2 BASE LEARNING

Unlike in Section 3.1, varying the base $\mathbb{G}$ does not need to preserve the same support. Thus, it is generally not possible to use maximum likelihood methods for learning $\mathbb{G}$. Instead, we propose to use the generalized likelihood (6) evaluated at the optimal energy $E^\star$ as a meaningful loss for learning $\mathbb{G}$, and refer to it as the *KL Approximate Lower-bound Estimate* (KALE),

$$\mathrm{KALE}(\mathbb{P}||\mathbb{G}) = \sup_{(E,A)\in\mathcal{E}\times\mathbb{R}} \mathcal{F}_{\mathbb{P},\mathbb{G}}(E+A). \tag{10}$$

From Section 3.1, $\mathrm{KALE}(\mathbb{P}||\mathbb{G})$ is always a lower bound on $\mathrm{KL}(\mathbb{P},\mathbb{G})$. The bound becomes tight whenever the negative log density of $\mathbb{P}$ w.r.t. $\mathbb{G}$ is well-defined and belongs to $\mathcal{E}$ (Appendix A). Moreover, Proposition 4 shows that KALE is a reliable criterion for measuring convergence, and is a consequence of (Zhang et al., 2017, Theorem B.1), with a proof in Appendix C.2.1:

**Proposition 4.** *Assume all energies in $\mathcal{E}$ are L-Lipschitz and that any continuous function can be well approximated by linear combinations of energies in $\mathcal{E}$ (Assumptions **(A)** and **(B)** of Appendix C.2), then KALE($\mathbb{P}||\mathbb{G}) \geq 0$ with equality only if $\mathbb{P} = \mathbb{G}$ and KALE($\mathbb{P}||\mathbb{G}^n) \to 0$ iff $\mathbb{G}^n \to \mathbb{P}$ in distribution.*

The universal approximation assumption holds in particular when $\mathcal{E}$ contains feedforward networks. In fact networks with a single neuron are enough, as shown in (Zhang et al., 2017, Theorem 2.3). The Lipschitz assumption holds when additional regularization of the energy is enforced during training by methods such as **spectral normalization** (Miyato et al., 2018) or additional regularization $I(\psi)$ on the energy $E_\psi$ such as the **gradient penalty** (Gulrajani et al., 2017) as done in Section 6.

**Estimating KALE.** According to Arora et al. (2017), accurate finite sample estimates of divergences that result from an optimization procedures (such as in (10)) depend on the richness of the class $\mathcal{E}$; and richer energy classes can result in slower convergence. Unlike divergences such as Jensen-Shannon, KL and the Wasserstein distance, which result from optimizing over a non-parametric and rich class of functions, KALE is restricted to a class of parametric energies $E_\psi$. Thus, (Arora et al., 2017, Theorem 3.1) applies, and guarantees good finite sample estimates, provided optimization is solved accurately. In Appendix B, we provide an analysis for the more general case where energies are not necessarily parametric but satisfy some further smoothness properties; we emphasize that our rates do not require the strong assumption that the density ratio is bounded above and below as in (Nguyen et al., 2010).

**Smoothness of KALE.** Learning the base is achieved by minimizing $\mathcal{K}(\theta) := \mathrm{KALE}(\mathbb{P}||\mathbb{G}_\theta)$ over the set of parameters $\Theta$ of the generator $G_\theta$ using first order methods (Duchi et al., 2011; Kingma and Ba, 2014; Arbel et al., 2019). This requires $\mathcal{K}(\theta)$ to be smooth enough so that gradient methods converge to local minima and avoid instabilities during training (Chu et al., 2020). Ensuring smoothness of losses that result from an optimization procedure, as in (10), can be challenging. Results for the regularized Wasserstein are provided by Sanjabi et al. (2018), while more general losses are considered by Chu et al. (2020), albeit under stronger conditions than for our setting. Theorem 5 shows that when $E$, $G_\theta$ and their gradients are all Lipschitz then $\mathcal{K}(\theta)$ is smooth enough. We provide a proof for Theorem 5 in Appendix C.2.1.

**Theorem 5.** *Under Assumptions (I) to (III) of Appendix C.2, sub-gradient methods on $\mathcal{K}$ converge to local optima. Moreover, $\mathcal{K}$ is Lipschitz and differentiable for almost all $\theta \in \Theta$ with:*

$$\nabla\mathcal{K}(\theta) = \exp(-A_{G_\theta, E^\star}) \int \nabla_x E^\star(G_\theta(z)) \nabla_\theta G_\theta(z) \exp(-E^\star(G_\theta(z))) \eta(z) \mathrm{d}z. \tag{11}$$

**Estimating the gradient** in (11) is achieved by first optimizing over $E_\psi$ and $A$ using (9), with additional regularization $I(\psi)$. The resulting estimators $\hat{E}^\star$ and $\hat{A}^\star$ are plugged in (12) to estimate $\nabla\mathcal{K}(\theta)$ using samples $(Z_m)_{1:M}$ from $\eta$. Unlike for learning the energy $E^\star$, which benefits from using the amortized estimator of the log-partition function, we found that using the empirical log-partition for learning the base was more stable. We summarize the training procedure in Algorithm 1, which alternates between learning the energy and the base in a similar fashion to *adversarial training*.

---

**Algorithm 1** Training GEBM

1: **Input** $\mathbb{P}$, $N$, $M$, $n_b$, $n_e$
2: **Output** Trained generator $G_\theta$ and energy $E_\psi$.
3: *Initialize $\theta$, $\psi$ and $A$.*
4: **for** $k = 1, ..., n_b$ **do**
5:     **for** $j = 1, ..., n_e$ **do**
6:         Sample $\{X_n\}_{1:N} \sim \mathbb{P}$ and $\{Y_n\}_{1:N} \sim \mathbb{G}_\theta$
7:         $g_\psi \leftarrow -\nabla_\psi \hat{\mathcal{F}}_{\mathbb{P}, \mathbb{G}_\theta}(E_\psi + A) + I(\psi)$
8:         $\tilde{A} \leftarrow \log\left(\frac{1}{M}\sum_{m=1}^{M}\exp(-E_\psi(Y_m))\right)$
9:         $g_A \leftarrow \exp(A - \tilde{A}) - 1$
10:         *Update $\psi$ and $A$ using $g_\psi$ and $g_A$.*
11:     **end for**
12:     Set $\hat{E}^\star \leftarrow E_\psi$ and $\hat{A}^\star \leftarrow A$.
13:     *Update $\theta$ using $\widehat{\nabla\mathcal{K}(\theta)}$ from (12)*
14: **end for**

---

$$\widehat{\nabla\mathcal{K}(\theta)} = \frac{\exp(-\hat{A}^\star)}{M} \sum_{m=1}^{M} \nabla_x \hat{E}^\star(G_\theta(Z_m)) \nabla_\theta G_\theta(Z_m) \exp(-\hat{E}^\star(G_\theta(Z_m))). \tag{12}$$

## 4   SAMPLING FROM GEBMS

A simple estimate of the empirical distribution of observations under the GEBM is via importance sampling (IS). This consists in first sampling multiple points from the base $\mathbb{G}$, and then re-weighting the samples according to the energy $E$. Although straightforward, this approach can lead to highly unreliable estimates, a well known problem in the Sequential Monte Carlo (SMC) literature which employs IS extensively (Doucet et al., 2001; Del Moral et al., 2006). Other methods such as rejection sampling are known to be inefficient in high dimensions Haugh (2017). Instead, we propose to sample from the posterior $\nu$ using MCMC. Recall from (5) that a sample $x$ from $\mathbb{Q}$ is of the form $x = G(z)$ with $z$ sampled from the *posterior latent $\nu$* of (4) instead of the prior $\eta$. While sampling from $\eta$ is often straightforward (for instance if $\eta$ is a Gaussian), sampling from $\nu$ is generally harder, due to dependence of its density on complex functions $E$ and $G$. It is still possible to use MCMC methods to sample from $\nu$, however, since we have access to its density up to a normalizing constant (4). In particular, we are interested in methods that exploit the gradient of $\nu$, and consider two classes of samplers: *Overdamped samplers* and *Kinetic samplers*.

**Overdamped samplers** are obtained as a time-discretization of the *Overdamped Langevin dynamics*:

$$dz_t = (\nabla_z\log\eta(z_t) - \nabla_z E(G(z_t))) + \sqrt{2}\mathrm{d}w_t, \tag{13}$$

where $w_t$ is a standard Brownian motion. The simplest sampler arising from (13) is the Unadjusted Langevin Algorithm (ULA):

$$Z_{k+1} = Z_k + \lambda(\nabla_z\log\eta(Z_k) - \nabla_z E(G(Z_k))) + \sqrt{2\lambda}W_{k+1}, \qquad Z_0 \sim \eta,$$

where $(W_k)_{k \geq 0}$ are i.i.d. standard Gaussians and $\lambda$ is the step-size. For large $k$, $Z_k$ is an approximate sample from $\nu$ (Raginsky et al., 2017, Proposition 3.3). Hence, setting $X = G(Z_k)$ for a large enough $k$ provides an approximate sample from the GEBM $\mathbb{Q}$, as summarized in Algorithm 2 of Appendix F.

**Kinetic samplers** arise from the *Kinetic Langevin dynamics* which introduce a momentum variable:

$$\mathrm{d}z_t = v_t\mathrm{d}t, \qquad \mathrm{d}v_t = -\gamma v_t\mathrm{d}t + u(\nabla\log\eta(z_t) - \nabla E(G(z_t)))\mathrm{d}t + \sqrt{2\gamma u}\mathrm{d}w_t. \tag{14}$$

with friction coefficient $\gamma \geq 0$, inverse mass $u \geq 0$, **momentum** vector $v_t$ and standard Brownian motion $w_t$. When the mass $u^{-1}$ becomes negligible compared to the friction coefficient $\gamma$, i.e. $u\gamma^{-2} \approx 0$, standard results show that (14) recovers the Overdamped dynamics (13). Discretization in time of (14)

leads to Kinetic samplers similar to Hamiltonian Monte Carlo (Cheng et al., 2017; Sachs et al., 2017). We consider a particular algorithm from Sachs et al. (2017) which we call Kinetic Langevin Algorithm (KLA) (see Algorithm 3 in Appendix F). Kinetic samplers were shown to better explore the modes of the invariant distribution $\nu$ compared to Overdamped ones (see (Neal, 2010; Betancourt et al., 2017) for empirical results and (Cheng et al., 2017) for theory), as also confirmed empirically in Appendix D for image generation tasks using GEBMs. Next, we provide the following convergence result:

**Proposition 6.** *Assume that $\log\eta(z)$ is strongly concave and has a Lipschitz gradient, that $E$, $G$ and their gradients are all $L$-Lipschitz. Set $x_t = G(z_t)$, where $z_t$ is given by (14) and call $\mathbb{P}_t$ the probability distribution of $x_t$. Then $\mathbb{P}_t$ converges to $\mathbb{Q}$ in the Wasserstein sense,*

$$W_2(\mathbb{P}_t, \mathbb{Q}) \le LCe^{-c\gamma t},$$

*where $c$ and $C$ are positive constants independent of $t$, with $c = O(\exp(-dim(\mathcal{Z})))$.*

Proposition 6 is proved in Appendix C.1 using (Eberle et al., 2017, Corollary 2.6), and implies that $(x_t)_{t \ge 0}$ converges at the same speed as $(z_t)_{t \ge 0}$. When the dimension $q$ of $\mathcal{Z}$ is orders of magnitude smaller than the input space dimension $d$, the process $(x_t)_{t \ge 0}$ converges faster than typical sampling methods on $\mathcal{X}$, for which the exponent controlling the convergence rate is of order $O(\exp(-d))$.

## 5 RELATED WORK

**Energy based models.** Usually, energy based models are required to have a density w.r.t. to a Lebesgue measure, and do not use a learnable base measure; in other words, models are supported on the whole space. Various methods have been proposed in the literature to learn EBMs. *Contrastive Divergence* (Hinton, 2002) approximates the gradient of the log-likelihood by sampling from the energy model with MCMC. More recently, (Belanger and McCallum, 2016; Xie et al., 2016; 2017; 2018c; 2019; Tu and Gimpel, 2018; Du and Mordatch, 2019; Deng et al., 2020) extend the idea using more sophisticated models and MCMC sampling strategies that lead to higher quality estimators. *Score Matching* (Hyvärinen, 2005) calculates an alternative objective (the *score*) to the log-likelihood which is independent of the partition function, and was recently used in the context non-parametric energy functions to provide estimators of the energy that are provably consistent (Sriperumbudur et al., 2017; Sutherland et al., 2018; Arbel and Gretton, 2018; Wenliang et al., 2019). In *Noise-Contrastive Estimation* (Gutmann and Hyvärinen, 2012), a classifier is trained to distinguish between samples from a fixed proposal distribution and the target $\mathbb{P}$. This provides an estimate for the density ratio between the optimal energy model and the proposal distribution. In a similar spirit, Cranmer et al. (2016) uses a classifier to learn likelihood ratios. Conversely, Grathwohl et al. (2020) interprets the logits of a classifier as an energy model obtained after marginalization over the classes. The resulting model is then trained using Contrastive Divergence. In more recent work, Dai et al. (2019a;b) exploit a dual formulation of the logarithm of the partition function as a supremum over the set of all probability distributions of some functional objective. Yu et al. (2020) explore methods for using general f-divergences, such as Jensen-Shannon, to train EBMs.

**Generative Adversarial Networks.** Recent work proposes using the discriminator of a trained GAN to improve the generator quality. Rejection sampling (Azadi et al., 2019) and Metropolis-Hastings correction (Turner et al., 2019; Neklyudov et al., 2019) perform sampling directly on the high-dimensional input space without using gradient information provided by the discriminator. Moreover, the data distribution is assumed to admit a density w.r.t. the generator. Ding et al. (2019) perform sampling on the feature space of some auxiliary pre-trained network; while Lawson et al. (2019) treat the sampling procedure as a model on its own, learned by maximizing the ELBO. In our case, no auxiliary model is needed. In the present work, sampling doesn't interfere with training, in contrast to recently considered methods to optimize over the latent space during training Wu et al. (2019b;a). In Tanaka (2019), the discriminator is viewed as an optimal transport map between the generator and the data distribution and is used to compute optimized samples from latent space. This is in contrast to the diffusion-based sampling that we consider. In (Xie et al., 2018b;a), two independent models, a full support EBM and a generator network, are trained cooperatively using MCMC. By contrast, in the present work, the energy and base are part of the same model, and the model support is lower-dimensional than the target space $\mathcal{X}$. While we do not address the mode collapse problem, Xu et al. (2018); Nguyen et al. (2017) showed that KL-based losses are resilient to it thanks to the zero-avoiding property of the KL, a good sign for KALE which is derived from KL by Fenchel duality.

The closest related approach appears in a study concurrent to the present work (Che et al., 2020), where the authors propose to use Langevin dynamics on the latent space of a GAN generator, but with a different discriminator to ours (derived from the Jensen-Shannon divergence or a Wasserstein-based divergence). Our theory results showing the existence of the loss gradient (Theorem 5), establishing weak convergence of distributions under KALE (Proposition 4), and demonstrating consistency of the KALE estimator (Appendix B) should transfer to the JS and Wasserstein criteria used in that work. Subsequent to the present work, an alternative approach has been recently proposed, based on normalising flows, to learn both the low-dimensional support of the data and the density on this support (Brehmer and Cranmer, 2020). This approach maximises the explicit likelihood of a data projection onto a learned manifold, and may be considered complementary to our approach.

## 6 EXPERIMENTS

### 6.1 IMAGE GENERATION.

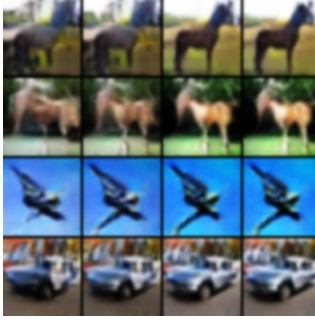

**Experimental setting.** We train a GEBM on **unsupervised** image generation tasks, and compare the quality of generated samples with other methods using the FID score (Heusel et al., 2017) computed on $5 \times 10^4$ generated samples. We consider CIFAR-10 (Krizhevsky, 2009), LSUN (Yu et al., 2015), CelebA (Liu et al., 2015) and ImageNet (Russakovsky et al., 2014) all downsampled to 32x32 resolution to reduce computational cost. We consider two network architectures for each of the base and energy, a smaller one (SNGAN ConvNet) and a larger one (SNGAN ResNet), both of which are from Miyato et al. (2018). For the base we used the SNGAN generator networks from Miyato et al. (2018) with a 100-dimensional Gaussian for the latent noise $\eta$. For the energy we used the SNGAN discriminator networks from Miyato et al. (2018). (Details of the networks in Appendix G.1).

Figure 2: Samples at different iterations of the MCMC chain of Algorithm 3 (left to right).

We train the models for 150000 generator iterations using Algorithm 1. After training is completed, we rescale the energy by $\beta = 100$ to get a **colder** version of the GEBM and sample from it using either Algorithm 2 (ULA) or Algorithm 3 (KLA) with parameters $(\gamma = 100, u = 1)$. This colder temperature leads to an improved FID score, and needs relatively few MCMC iterations, as shown in Figure 6 of Appendix D. Sampler convergence to visually plausible modes at low tempteratures is demonstrated in Figure 2. We perform 1000 MCMC iterations with initial step-size of $\lambda = 10^{-4}$ decreased by 10 every 200 iterations. As a baseline we consider samples generated from the base of the GEBM only (without using information from the energy) and call this KALE-GAN. More details are given in Appendix G.

**Results:** Table 1 shows that GEBM outperforms both KALE and standard GANs when using the same networks for the base/generator and energy/critic. Moreover, KALE-GAN matches the performance of a standard GAN (with Jensen-Shannon critic), showing that the improvement of GEBM cannot be explained by the switch from Jensen-Shannon to a KALE-based critic. Rather, the improvement is largely due to incorporating the energy function into the model, and sampling using Algorithm 3.

This finding experimentally validates our claim that incorporating the energy improves the model, and that all else being equal, a GEBM outperforms a GAN with the same generator and critic architecture. Indeed, if the critic is not zero at convergence, then by definition it contains information on the remaining mismatch between the generator (base) and data mass, which the GEBM incorporates, but the GAN does not. The GEBM also outperforms an EBM even when the latter was trained using a larger network (ResNet) with supervision (S) on ImageNet, which is an easier task ( Chen et al. (2019)). More comparisons on Cifar10 and ImageNet are provided in Table 4 of Appendix D.

| | SNGAN (ConvNet) | | | SNGAN (ResNet) | | | |
|---|---|---|---|---|---|---|---|
| | GEBM | KALE-GAN | GAN | GEBM | KALE-GAN | GAN | EBM |
| Cifar10 | 23.02 | 32.03 | 29.9 | **19.31** | 20.19 | 21.7 | 38.2 |
| ImageNet | **13.94** | 19.37 | 20.66 | 20.33 | 21.00 | 20.50 | 14.31 (S) |

Table 1: FID scores for two versions of SNGAN from (Miyato et al., 2018) on Cifar10 and ImageNet. GEBM: training using Algorithm 1 and sampling using Algorithm 3. KALE-GAN: Only the base of a GEBM is retained for sampling. GAN: training as in (Miyato et al., 2018) with $q = 128$ for the latent dimension as it worked best. EBM: results from Du and Mordatch (2019) with *supervised* training on ImageNet (S).

Table 2 shows different sampling methods using the same trained networks (generator and critic), with KALE-GAN as a baseline. All energy-exploiting methods outperform the unmodified KALE-GAN with the same architecture. That said, our method (both ULA and KLA) outperforms both (IHM) (Turner et al., 2019) and (DOT) (Tanaka, 2019), which both use the energy information.

|  | Cifar10 | LSUN | CelebA | ImageNet |
|---|---|---|---|---|
| KALE-GAN | 32.03 | 21.67 | 6.91 | 19.37 |
| IHM | 30.47 | 20.63 | 6.39 | 18.15 |
| DOT | 26.35 | 20.41 | 5.93 | 16.21 |
| GEBM (ULA) | **23.02** | 16.23 | **5.21** | 14.00 |
| GEBM (KLA) | 24.29 | **15.25** | 5.38 | **13.94** |

Table 2: FID scores for different sampling methods using the same trained SNGAN (ConvNet): KALE-GAN as a baseline w/o critic information.

In Table 2, KLA was used in the high friction regime $\gamma = 100$ and thus behaves like ULA. This allows to obtain sharper samples concentrated around the modes of the GEBM thus improving the FID score. If, instead, the goal is to encourage more exploration of the modes of the GEBM, then KLA with a smaller $\gamma$ is a better alternative than ULA, as the former can explore multiple modes/images within the same MCMC chain, unlike (ULA): see Figures 3 to 5 of Appendix D. Moving from one mode to another results in an increased FID score while between modes, however, which can be avoided by decreasing $\lambda$.

## 6.2 DENSITY ESTIMATION

**Motivation.** We next consider the particular setting where the likelihood of the model is well-defined, and admits a closed form expression. This is intended principally as a sanity check that our proposed training method in Algorithm 1 succeeds in learning maximum likelihood solutions. Outside of this setting, closed form expressions of the normalizing constant are not available for generic GEBMs. While this is not an issue (since the proposed method doesn't require a closed form expression for the normalizing constant), in this experiment only, we want to have access to closed form expressions, as they enable a direct comparison with other density estimation methods.

**Experimental setting.** To have a closed-form likelihood, we consider the case where the dimension of the latent space is equal to data-dimension, and choose the base $\mathbb{G}$ of the GEBM to be a Real NVP (Ding et al. (2019) ) with density $\exp(-r(x))$ and energy $E(x) = h(x) - r(x)$. Thus, in this particular case, the GEBM has a well defined likelihood over the whole space, and we are precisely in the setting of Proposition 2, which shows that this GEBM is equal to an EBM with density proportional to $exp(-h)$. We further require the EBM to be a second Real NVP so that its density has a closed form expression. We consider 5 UCI datasets (Dheeru and Taniskidou, 2017) for which we use the same pre-processing as in (Wenliang et al., 2019). For comparison, we train the EBM by direct maximum likelihood (ML) and contrastive divergence (CD). To train the GEBM, we use Algorithm 1, which doesn't directly exploit the closed-form expression of the likelihood (unlike direct ML). We thus use either (8) (KALE-DV) or (9) (KALE-F) to estimate the normalizing constant. More details are given in Appendix G.2.

**Results.** Table 3 reports the Negative Log-Likelihood (NLL) evaluated on the test set and corresponding to the best performance on the validation set. Training the GEBM using Algorithm 1 leads to comparable performance to (CD) and (ML). As shown in Figure 7 of Appendix E, (KALE-DV) and (KALE-F) maintain a small error gap between the training and test NLL and, as discussed in Section 3.1 and Appendix F, (KALE-F) leads to more accurate estimates of the log-partition function, with a relative error of order $0.1\%$ compared to $10\%$ for (KALE-DV).

|  | RedWine $d=11, N \sim 10^3$ | Whitewine $d=11, N \sim 10^3$ | Parkinsons $d=15, N \sim 10^3$ | Hepmass $d=22, N \sim 10^5$ | Miniboone $d=43, N \sim 10^4$ |
|---|---|---|---|---|---|
| **NVP w ML** | 11.98 | 13.05 | 14.5 | 24.89 | 42.28 |
| **NVP w CD** | 11.88 | 13.01 | 14.06 | **22.89** | 39.36 |
| **NVP w KALE (DV)** | 11.6 | 12.77 | **13.26** | 26.56 | 46.48 |
| **NVP w KALE (F)** | **11.19** | **12.66** | **13.26** | 24.66 | **38.35** |

Table 3: UCI datasets: Negative log-likelihood computed on the test set and corresponding to the best performance on the validation set. Best method in boldface.

## 7 ACKNOWLEDGMENTS

We thank Mihaela Rosca for insightful discussions and Song Liu, Bo Dai and Hanjun Dai for pointing us to important related work.

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

## A    KL Approximate Lower-bound Estimate

We discuss the relation between KALE (10) and the Kullback-Leibler divergence via Fenchel duality. Recall that a distribution $\mathbb{P}$ is said to admit a density w.r.t. $\mathbb{G}$ if there exists a real-valued measurable function $r_0$ that is integrable w.r.t. $\mathbb{G}$ and satisfies $d\mathbb{P} = r_0 d\mathbb{G}$. Such a density is also called the *Radon-Nikodym derivative* of $\mathbb{P}$ w.r.t. $\mathbb{G}$. In this case, we have:

$$\mathrm{KL}(\mathbb{P}||\mathbb{G}) = \int r_0 \log(r_0) d\mathbb{G}. \tag{15}$$

Nguyen et al. (2010); Nowozin et al. (2016) derived a variational formulation for the KL using Fenchel duality. By the duality theorem (Rockafellar, 1970), the convex and lower semi-continuous function $\zeta : u \mapsto u \log(u)$ that appears in (15) can be expressed as the supremum of a concave function:

$$\zeta(u) = \sup_v uv - \zeta^\star(v).$$

The function $\zeta^\star$ is called the *Fenchel dual* and is defined as $\zeta^\star(v) = \sup_u uv - \zeta(u)$. By convention, the value of the objective is set to $-\infty$ whenever $u$ is outside of the domain of definition of $\zeta^\star$. When $\zeta(u) = u \log(u)$, the Fenchel dual $\zeta^\star(v)$ admits a closed form expression of the form $\zeta^\star(v) = \exp(v-1)$. Using the expression of $\zeta$ in terms of its Fenchel dual $\zeta^\star$, it is possible to express $\mathrm{KL}(\mathbb{P}||\mathbb{G})$ as the supremum of the variational objective (16) over all measurable functions $h$.

$$\mathcal{F}(h) := -\int h \, d\mathbb{P} - \int \exp(-h) d\mathbb{G} + 1. \tag{16}$$

Nguyen et al. (2010) provided the variational formulation for the reverse KL using a different choice for $\zeta$: ($\zeta(u) = -\log(u)$). We refer to (Nowozin et al., 2016) for general $f$-divergences. Choosing a smaller set of functions $\mathcal{H}$ in the variational objective (16) will lead to a lower bound on the KL. This is the *KL Approximate Lower-bound Estimate* (KALE):

$$\mathrm{KALE}(\mathbb{P}||\mathbb{G}) = \sup_{h \in \mathcal{H}} \mathcal{F}(h) \tag{17}$$

In general, $\mathrm{KL}(\mathbb{P}||\mathbb{G}) \geq \mathrm{KALE}(\mathbb{P}||\mathbb{G})$. The bound is tight whenever the negative log-density $h_0 = -\log r_0$ belongs to $\mathcal{H}$; however, we do not require $r_0$ to be well-defined in general. Equation (17) has the advantage that it can be estimated using samples from $\mathbb{P}$ and $\mathbb{G}$. Given i.i.d. samples $(X_1, ..., X_N)$ and $(Y_1, ..., Y_M)$ from $\mathbb{P}$ and $\mathbb{G}$, we denote by $\hat{\mathbb{P}}$ and $\hat{\mathbb{G}}$ the corresponding empirical distributions. A simple approach to estimate $\mathrm{KALE}(\mathbb{P}||\mathbb{G})$ is to use an $M$-estimator. This is achieved by optimizing the penalized objective

$$\hat{h} := \arg\max_{h \in \mathcal{H}} \widehat{\mathcal{F}}(h) - \frac{\lambda}{2} I^2(h), \tag{18}$$

where $\widehat{\mathcal{F}}$ is an empirical version of $\mathcal{F}$ and $I^2(h)$ is a penalty term that prevents overfitting due to finite samples. The penalty $I^2(h)$ acts as a regularizer favoring smoother solutions while the parameter $\lambda$ determines the strength of the smoothing and is chosen to decrease as the sample size $N$ and $M$ increase. The $M$-estimator of $\mathrm{KALE}(\mathbb{P}||\mathbb{G})$ is obtained simply by plugging in $\hat{h}$ into the empirical objective $\widehat{\mathcal{F}}(h)$:

$$\widehat{\mathrm{KALE}}(\mathbb{P}||\mathbb{G}) := \widehat{\mathcal{F}}(\hat{h}). \tag{19}$$

We defer the consistency analysis of (19) to Appendix B where we provide convergence rates in a setting where the set of functions $\mathcal{H}$ is a Reproducing Kernel Hilbert Space and under weaker assumptions that were not covered by the framework of Nguyen et al. (2010).

## B    Convergence rates of KALE

In this section, we provide a convergence rate for the estimator in (19) when $\mathcal{H}$ is an RKHS. The theory remains the same whether $\mathcal{H}$ contains constants or not. With this choice, the Representer Theorem allows us to reduce the potentially infinite-dimensional optimization problem in (18) to a convex finite-dimensional one. We further restrict ourselves to the *well-specified* case where the density $r_0$ of $\mathbb{P}$ w.r.t. $\mathbb{G}$ is well-defined and belongs to $\mathcal{H}$, so that KALE matches the KL. While Nguyen et al.

(2010) (Theorem 3) provides a convergence rate of $1/\sqrt{N}$ for a related $M$-estimator, this requires the density $r_0$ to be lower-bounded by 0 as well as (generally) upper-bounded. This can be quite restrictive if, for instance, $r_0$ is the density ratio of two gaussians. In Theorem 7, we provide a similar convergence rate for the estimator defined in (19) without requiring $r_0$ to be bounded. We start by briefly introducing some notations, the working assumptions and the statement of the convergence result in Appendix B.1 and provide the proofs in Appendix B.2.

### B.1 STATEMENT OF THE RESULT

We recall that an RKHS $\mathcal{H}$ of functions defined on a domain $\mathcal{X} \subset \mathbb{R}^d$ and with kernel $k$ is a Hilbert space with dot product $\langle .,. \rangle$, such that $y \mapsto k(x,y)$ belongs to $\mathcal{H}$ for any $x \in \mathcal{X}$, and

$$k(x,y) = \langle k(x,.), k(y,.) \rangle, \qquad \forall x, y \in \mathcal{X}.$$

Any function $h$ in $\mathcal{H}$ satisfies the reproducing property $f(x) = \langle f, k(x,.) \rangle$ for any $x \in \mathcal{X}$.

Recall that $\mathrm{KALE}(\mathbb{P}||\mathbb{G})$ is obtained as an optimization problem

$$\mathrm{KALE}(\mathbb{P}||\mathbb{G}) = \sup_{h \in \mathcal{H}} \mathcal{F}(h) \tag{20}$$

where $\mathcal{F}$ is given by:

$$\mathcal{F}(h) := -\int h\,d\mathbb{P} - \int \exp(-h)\,d\mathbb{G} + 1.$$

Since the negative log density ratio $h_0$ is assumed to belong to $\mathcal{H}$, this directly implies that the supremum of $\mathcal{F}$ is achieved at $h_0$ and $\mathcal{F}(h_0) = \mathrm{KALE}(\mathbb{P}||\mathbb{G})$. We are interested in estimating $\mathrm{KALE}(\mathbb{P}||\mathbb{G})$ using the empirical distributions $\hat{\mathbb{P}}$ and $\hat{\mathbb{G}}$,

$$\hat{\mathbb{P}} := \frac{1}{N} \sum_{n=1}^{N} \delta_{X_n}, \qquad \hat{\mathbb{G}} := \frac{1}{N} \sum_{n=1}^{N} \delta_{Y_n},$$

where $(X_n)_{1 \leq n \leq N}$ and $(Y_n)_{1 \leq n \leq N}$ are i.i.d. samples from $\mathbb{P}$ and $\mathbb{G}$. For this purpose we introduce the empirical objective functional,

$$\widehat{\mathcal{F}}(h) := -\int h\,d\hat{\mathbb{P}} - \int \exp(-h)\,d\hat{\mathbb{G}} + 1.$$

The proposed estimator is obtained by solving a regularized empirical problem,

$$\sup_{h \in \mathcal{H}} \widehat{\mathcal{F}}(h) - \frac{\lambda}{2}\|h\|^2, \tag{21}$$

with a corresponding population version,

$$\sup_{h \in \mathcal{H}} \mathcal{F}(h) - \frac{\lambda}{2}\|h\|^2. \tag{22}$$

Finally, we introduce $D(h,\delta)$ and $\Gamma(h,\delta)$:

$$D(h,\delta) = \int \delta \exp(-h)\,d\mathbb{G} - \int \delta\,d\mathbb{P},$$

$$\Gamma(h,\delta) = -\int\int_0^1 (1-t)\delta^2 \exp(-(h+t\delta))\,d\mathbb{G}.$$

The empirical versions of $D(h,\delta)$ and $\Gamma(h,\delta)$ are denoted $\hat{D}(h,\delta)$ and $\hat{\Gamma}(h,\delta)$. Later, we will show that $D(h,\delta)$ $\hat{D}(h,\delta)$ are in fact the gradients of $\mathcal{F}(h)$ and $\widehat{\mathcal{F}}(h)$ along the direction $\delta$.

We state now the working assumptions:

    (**i**) The supremum of $\mathcal{F}$ over $\mathcal{H}$ is attained at $h_0$.

(ii) The following quantities are finite for some positive $\epsilon$:

$$\int \sqrt{k(x,x)}\,d\mathbb{P}(x),$$

$$\int \sqrt{k(x,x)}\exp((\|h_0\|+\epsilon)\sqrt{k(x,x)})\,d\mathbb{G}(x),$$

$$\int k(x,x)\exp((\|h_0\|+\epsilon)\sqrt{k(x,x)})\,d\mathbb{G}(x).$$

(iii) For any $h \in \mathcal{H}$, if $D(h,\delta)=0$ for all $\delta$ then $h=h_0$.

**Theorem 7.** *Fix any $1 > \eta > 0$. Under Assumptions (i) to (iii), and provided that $\lambda = \frac{1}{\sqrt{N}}$, it holds with probability at least $1-2\eta$ that*

$$|\widehat{\mathcal{F}}(\hat{h}) - \mathcal{F}(h_0)| \leq \frac{M'(\eta,h_0)}{\sqrt{N}}$$

*for a constant $M'(\eta,h_0)$ that depends only on $\eta$ and $h_0$.*

The assumptions in Theorem 7 essentially state that the kernel associated to the RKHS $\mathcal{H}$ needs to satisfy some integrability requirements. That is to guarantee that the gradient $\delta \mapsto \nabla\mathcal{F}(h)(\delta)$ and its empirical version are well-defined and continuous. In addition, the optimality condition $\nabla\mathcal{F}(h)=0$ is assumed to characterize the global solution $h_0$. This will be the case if the kernel is characteristic Simon-Gabriel and Scholkopf (2018). The proof of Theorem 7, in Appendix B.2, takes advantage of the Hilbert structure of the set $\mathcal{H}$, the convexity of the functional $\mathcal{F}$ and the optimality condition $\nabla\widehat{\mathcal{F}}(\hat{h})=\lambda\hat{h}$ of the regularized problem, all of which turn out to be sufficient for controlling the error of (19).

## B.2 PROOFS

We state now the proof of Theorem 7 with subsequent lemmas and propositions.

*Proof of Theorem 7.* We begin with the following inequalities:

$$\frac{\lambda}{2}(\|\hat{h}\|^2 - \|h_0\|^2) \leq \widehat{\mathcal{F}}(\hat{h}) - \widehat{\mathcal{F}}(h_0) \leq \langle \nabla\widehat{\mathcal{F}}(h_0), \hat{h}-h_0\rangle.$$

The first inequality is by definition of $\hat{h}$ while the second is obtained by concavity of $\widehat{\mathcal{F}}$. For simplicity we write $\mathcal{B}=\|\hat{h}-h_0\|$ and $\mathcal{C}=\|\nabla\widehat{\mathcal{F}}(h_0)-\mathcal{L}(h_0)\|$. Using Cauchy-Schwarz and triangular inequalities, it is easy to see that

$$-\frac{\lambda}{2}(\mathcal{B}^2+2\mathcal{B}\|h_0\|) \leq \widehat{\mathcal{F}}(\hat{h}) - \widehat{\mathcal{F}}(h_0) \leq \mathcal{C}\mathcal{B}.$$

Moreover, by triangular inequality, it holds that

$$\mathcal{B} \leq \|h_\lambda - h_0\| + \|\hat{h}-h_\lambda\|.$$

Lemma 11 ensures that $\mathcal{A}(\lambda)=\|h_\lambda - h_0\|$ converges to 0 as $\lambda \to 0$. Furthermore, by Proposition 12, we have $\|\hat{h}-h_\lambda\| \leq \frac{1}{\lambda}\mathcal{D}$ where $\mathcal{D}(\lambda)=\|\nabla\widehat{\mathcal{F}}(h_\lambda)-\nabla\mathcal{L}(h_\lambda)\|$. Now choosing $\lambda=\frac{1}{\sqrt{N}}$ and applying Chebychev inequality in Lemma 8, it follows that for any $1 > \eta > 0$, we have with probability greater than $1-2\eta$ that both

$$\mathcal{D}(\lambda) \leq \frac{C(\|h_0\|,\eta)}{\sqrt{N}}, \qquad \mathcal{C} \leq \frac{C(\|h_0\|,\eta)}{\sqrt{N}},$$

where $C(\|h_0\|,\eta)$ is defined in Lemma 8. This allows to conclude that for any $\eta > 0$, it holds with probability at least $1-2\eta$ that $|\widehat{\mathcal{F}}(\hat{h})-\widehat{\mathcal{F}}(h_0)| \leq \frac{M'(\eta,h_0)}{\sqrt{N}}$ where $M'(\eta,h_0)$ depends only on $\eta$ and $h_0$.

$\square$

We proceed using the following lemma, which provides an expression for $D(h,\delta)$ and $\hat{D}(h,\delta)$ along with a probabilistic bound:

**Lemma 8.** *Under Assumptions (i) and (ii), for any $h \in \mathcal{H}$ such that $\|h\| \leq \|h_0\| + \epsilon$, there exists $\mathcal{D}(h)$ in $\mathcal{H}$ satisfying*

$$D(h, \delta) = \langle \delta, \mathcal{D}(h) \rangle,$$

*and for any $h \in \mathcal{H}$, there exists $\widehat{\mathcal{D}}(h)$ satisfying*

$$\widehat{D}(h, \delta) = \langle \delta, \widehat{\mathcal{D}}(h) \rangle.$$

*Moreover, for any $0 < \eta < 1$ and any $h \in \mathcal{H}$ such that $\|h\| \leq \|h_0\| + \epsilon := M$, it holds with probability greater than $1 - \eta$ that*

$$\|\mathcal{D}(h) - \widehat{\mathcal{D}}(h)\| \leq \frac{C(M, \eta)}{\sqrt{N}},$$

*where $C(M, \eta)$ depends only on $M$ and $\eta$.*

*Proof.* First, we show that $\delta \mapsto D(h, \delta)$ is a bounded linear operator. Indeed, Assumption (ii) ensures that $k(x, .)$ and $k(x, .)\exp(-h(x))$ are Bochner integrable w.r.t. $\mathbb{P}$ and $\mathbb{G}$ (Retherford (1978)), hence $D(h, \delta)$ is obtained as

$$D(h, \delta) := \langle \delta, \mu_{\exp(-h)\mathbb{G}} - \mu_{\mathbb{P}} \rangle,$$

where $\mu_{\exp(-h)\mathbb{G}} = \int k(x, .)\exp(-h(x)) d\mathbb{G}$ and $\mu_{\mathbb{P}} = \int k(x, .) d\mathbb{P}$. Defining $\mathcal{D}(h)$ to be $= \mu_{\exp(-h)\mathbb{G}} - \mu_{\mathbb{P}}$ leads to the desired result. $\widehat{\mathcal{D}}(h)$ is simply obtained by taking the empirical version of $\mathcal{D}(h)$.

Finally, the probabilistic inequality is a simple consequence of Chebychev's inequality. $\square$

The next lemma states that $\mathcal{F}(h)$ and $\widehat{\mathcal{F}}(h)$ are Frechet differentiable.

**Lemma 9.** *Under Assumptions (i) and (ii), $h \mapsto \mathcal{F}(h)$ is Frechet differentiable on the open ball of radius $\|h_0\| + \epsilon$ while $h \mapsto \widehat{\mathcal{F}}(h)$ is Frechet differentiable on $\mathcal{H}$. Their gradients are given by $\mathcal{D}(h)$ and $\widehat{\mathcal{D}}(h)$ as defined in Lemma 8,*

$$\nabla \mathcal{F}(h) = \mathcal{D}(h), \qquad \nabla \widehat{\mathcal{F}}(h) = \widehat{\mathcal{D}}(h)$$

*Proof.* The empirical functional $\widehat{\mathcal{F}}(h)$ is differentiable since it is a finite sum of differentiable functions, and its gradient is simply given by $\widehat{\mathcal{D}}(h)$. For the population functional, we use second order Taylor expansion of $\exp$ with integral remainder, which gives

$$\mathcal{F}(h + \delta) = \mathcal{F}(h) - D(h, \delta) + \Gamma(h, \delta).$$

By Assumption (ii) we know that $\frac{\Gamma(h, \delta)}{\|\delta\|}$ converges to 0 as soon as $\|\delta\| \to 0$. This allows to directly conclude that $\mathcal{F}$ is Frechet differentiable, with differential given by $\delta \mapsto D(h, \delta)$. By Lemma 8, we conclude the existence of a gradient $\nabla \mathcal{F}(h)$ which is in fact given by $\nabla \mathcal{F}(h) = \mathcal{D}(h)$.

$\square$

From now on, we will only use the notation $\nabla \mathcal{F}(h)$ and $\nabla \widehat{\mathcal{F}}(h)$ to refer to the gradients of $\mathcal{F}(h)$ and $\widehat{\mathcal{F}}(h)$. The following lemma states that (21) and (22) have a unique global optimum, and gives a first order optimality condition.

**Lemma 10.** *The problems (21) and (22) admit unique global solutions $\hat{h}$ and $h_\lambda$ in $\mathcal{H}$. Moreover, the following first order optimality conditions hold:*

$$\lambda \hat{h} = \nabla \widehat{\mathcal{F}}(\hat{h}), \qquad \lambda h_\lambda = \nabla \mathcal{F}(h_\lambda).$$

*Proof.* For (21), existence and uniqueness of a minimizer $\hat{h}$ is a simple consequence of continuity and strong concavity of the regularized objective. We now show the existence result for (22). Let's introduce $\mathcal{G}_\lambda(h) = -\mathcal{F}(h) + \frac{\lambda}{2}\|h\|^2$ for simplicity. Uniqueness is a consequence of the strong convexity of $\mathcal{G}_\lambda$. For the existence, consider a sequence of elements $f_k \in \mathcal{H}$ such that $\mathcal{G}_\lambda(f_k) \to \inf_{h \in \mathcal{H}} \mathcal{G}_\lambda(h)$. If $h_0$

is not the global solution, then it must hold for $k$ large enough that $\mathcal{G}_\lambda(f_k) \leq \mathcal{G}_\lambda(h_0)$. We also know that $\mathcal{F}(f_k) \leq \mathcal{F}(h_0)$, hence, it is easy to see that $\|f_k\| \leq \|h_0\|$ for $k$ large enough. This implies that $f_k$ is a bounded sequence, therefore it admits a weakly convergent sub-sequence by weak compactness. Without loss of generality we assume that $f_k$ weakly converges to some element $h_\lambda \in \mathcal{H}$ and that $\|f_k\| \leq \|h_0\|$. Hence, $\|h_\lambda\| \leq \liminf_k \|f_k\| \leq \|h_0\|$. Recall now that by definition of weak convergence, we have $f_k(x) \to_k h_\lambda(x)$ for all $x \in \mathcal{X}$. By Assumption (ii), we can apply the dominated convergence theorem to ensure that $\mathcal{F}(f_k) \to \mathcal{F}(h_\lambda)$. Taking the limit of $\mathcal{G}_\lambda f_k$, the following inequality holds:

$$\sup_{h \in \mathcal{H}} \mathcal{G}_\lambda(h) = \limsup_k \mathcal{G}_\lambda(f_k) \leq \mathcal{G}_\lambda(h_\lambda).$$

Finally, by Lemma 9 we know that $\mathcal{F}$ is Frechet differentiable, hence we can use Ekeland and Témam (1999) (Proposition 2.1) to conclude that $\nabla \mathcal{F}(h_\lambda) = \lambda h_\lambda$. We use exactly the same arguments for (21). □

Next, we show that $h_\lambda$ converges towards $h_0$ in $\mathcal{H}$.

**Lemma 11.** *Under Assumptions (i) to (iii) it holds that:*

$$\mathcal{A}(\lambda) := \|h_\lambda - h_0\| \to 0.$$

*Proof.* We will first prove that $h_\lambda$ converges weakly towards $h_0$, and then conclude that it must also converge strongly. We start with the following inequalities:

$$0 \geq \mathcal{F}(h_\lambda) - \mathcal{F}(h_0) \geq \frac{\lambda}{2}(\|h_\lambda\|^2 - \|h_0\|^2).$$

These are simple consequences of the definitions of $h_\lambda$ and $h_0$ as optimal solutions to (20) and (21). This implies that $\|h_\lambda\|$ is always bounded by $\|h_0\|$. Consider now an arbitrary sequence $(\lambda_m)_{m \geq 0}$ converging to $0$. Since $\|h_{\lambda_m}\|$ is bounded by $\|h_0\|$, it follows by weak-compactness of balls in $\mathcal{H}$ that $h_{\lambda_m}$ admits a weakly convergent sub-sequence. Without loss of generality we can assume that $h_{\lambda_m}$ is itself weakly converging towards an element $h^*$. We will show now that $h^*$ must be equal to $h_0$. Indeed, by optimality of $h_{\lambda_m}$, it must hold that

$$\lambda_m h_{\lambda_m} = \nabla \mathcal{F}(h_m).$$

This implies that $\nabla \mathcal{F}(h_m)$ converges weakly to $0$. On the other hand, by Assumption (ii), we can conclude that $\nabla \mathcal{F}(h_m)$ must also converge weakly towards $\nabla \mathcal{F}(h^*)$, hence $\nabla \mathcal{F}(h^*) = 0$. Finally by Assumption (iii) we know that $h_0$ is the unique solution to the equation $\nabla \mathcal{F}(h) = 0$, hence $h^* = h_0$. We have shown so far that any subsequence of $h_{\lambda_m}$ that converges weakly, must converge weakly towards $h_0$. This allows to conclude that $h_{\lambda_m}$ actually converges weakly towards $h_0$. Moreover, we also have by definition of weak convergence that:

$$\|h_0\| \leq \lim_{m \to \infty} \inf \|h_{\lambda_m}\|.$$

Recalling now that $\|h_{\lambda_m}\| \leq \|h_0\|$ it follows that $\|h_{\lambda_m}\|$ converges towards $\|h_0\|$. Hence, we have the following two properties:

- $h_{\lambda_m}$ converges weakly towards $h_0$,

- $\|h_{\lambda_m}\|$ converges towards $\|h_0\|$.

This allows to directly conclude that $\|h_{\lambda_m} - h_0\|$ converges to $0$. □

**Proposition 12.** *We have that:*

$$\|\hat{h} - h_\lambda\| \leq \frac{1}{\lambda}\|\nabla \hat{\mathcal{F}}(h_\lambda) - \nabla \mathcal{F}(h_\lambda)\|$$

*Proof.* By definition of $\hat{h}$ and $h_\lambda$ the following optimality conditions hold:

$$\lambda \hat{h} = \nabla \widehat{\mathcal{F}}(\hat{h}), \qquad \lambda h_\lambda = \nabla \mathcal{F}(h_\lambda).$$

We can then simply write:

$$\lambda(\hat{h}-h_\lambda)-(\nabla\widehat{\mathcal{F}}(\hat{h})-\nabla\widehat{\mathcal{F}}(h_\lambda))=\nabla\widehat{\mathcal{F}}(h_\lambda)-\nabla\mathcal{F}(h_\lambda).$$

Now introducing $\delta:=\hat{h}-h_\lambda$ and $E:=\nabla\widehat{\mathcal{F}}(\hat{h})-\nabla\widehat{\mathcal{F}}(h_\lambda)$ for simplicity and taking the squared norm of the above equation, it follows that

$$\lambda^2\|\delta\|^2+\|E\|^2-2\lambda\langle\delta,E\rangle=\|\nabla\widehat{\mathcal{F}}(h_\lambda)-\nabla\mathcal{F}(h_\lambda)\|^2.$$

By concavity of $\widehat{\mathcal{F}}$ on $\mathcal{H}$ we know that $-\langle\hat{h}-h_\lambda,E\rangle\geq0$. Therefore:

$$\lambda^2\|\hat{h}-h_\lambda\|^2\leq\|\nabla\widehat{\mathcal{F}}(h_\lambda)-\nabla\mathcal{F}(h_\lambda)\|^2.$$

$\square$

# C  LATENT NOISE SAMPLING AND SMOOTHNESS OF KALE

## C.1  LATENT SPACE SAMPLING

Here we prove Proposition 6 for which we make the assumptions more precise:

**Assumption 1.** *We make the following assumption:*

- *$\log\eta$ is strongly concave and admits a Lipschitz gradient.*

- *There exists a non-negative constant $L$ such that for any $x,x'\in\mathcal{X}$ and $z,z'\in\mathcal{Z}$:*

$$|E(x)-E(x')|\leq\|x-x'\|, \qquad \|\nabla_x E(x)-\nabla_x E(x')\|\leq\|x-x'\|$$
$$|G(z)-G(z')|\leq\|z-z'\|, \qquad \|\nabla_z G(z)-\nabla_z G(z')\|\leq\|z-z'\|$$

Throughout this section, we introduce $U(z):=-\log(\eta(z))+E(G(z))$ for simplicity.

*Proof of Proposition 1 .*  To sample from $\mathbb{Q}_{\mathbb{G},E}$, we first need to identify the *posterior latent* distribution $\nu_{\mathbb{G},E}$ used to produce those samples. We rely on (23) which holds by definition of $\mathbb{Q}_{\mathbb{G},E}$ for any test function $h$ on $\mathcal{X}$:

$$\int h(x)\mathrm{d}\mathbb{Q}(x)=\int h(G(z))f(G(z))\eta(z)\mathrm{d}z, \tag{23}$$

Hence, the posterior latent distribution is given by $\nu(z)=\eta(z)f(G(z))$, and samples from GEBM are produced by first sampling from $\nu_{\mathbb{G},E}$, then applying the implicit map $G$,

$$X\sim\mathbb{Q}\quad\Longleftrightarrow\quad X=G(Z),\quad Z\sim\nu.$$

$\square$

*Proof of Proposition 2.*  the base distribution $\mathbb{G}$ admits a density on the whole space denoted by $\exp(-r(x))$ and the energy $\tilde{E}$ is of the form $\tilde{E}(x)=E(x)-r(x)$ for some parametric function $E$, it is easy to see that $\mathbb{Q}$ has a density proportional to $\exp(-E)$ and is therefore equivalent to a standard EBM with energy $E$.

The converse holds as well, meaning that for any EBM with energy $E$, it is possible to construct a GEBM using an *importance weighting* strategy. This is achieved by first choosing a base $\mathbb{G}$, which is required to have an explicit density $\exp(-r)$ up to a normalizing constant, then defining the energy of the GEBM to be $\tilde{E}(x)=E(x)-r(x)$ so that:

$$\mathrm{d}\mathbb{Q}(x)\propto\exp(-\tilde{E}(x))\mathrm{d}\mathbb{G}_\theta(x)\propto\exp(-E(x))\mathrm{d}x \tag{24}$$

Equation (24) effectively depends only on $E(x)$ and not on $\mathbb{G}$ since the factor $\exp(r)$ exactly compensates for the density of $\mathbb{G}$. The requirement that the base also admits a tractable implicit map $G$ can be met by choosing $\mathbb{G}$ to be a *normalizing flow* (Rezende and Mohamed, 2015) and does not restrict the class of possible EBMs that can be expressed as GEBMs.  $\square$

*Proof of Proposition 6.* Let $\pi_t$ be the probability distribution of $(z_t, v_t)$ at time $t$ of the diffusion in (14), which we recall that

$$dz_t = v_t dt, \qquad dv_t = -(\gamma v_t + u \nabla U(z_t)) + \sqrt{2\lambda u} dw_t,$$

We call $\pi_\infty$ its corresponding invariant distribution given by

$$\pi_\infty(z, v) \propto \exp\left(-U(z) - \frac{1}{2}\|v\|^2\right)$$

By Lemma 13 we know that $U$ is dissipative, bounded from below, and has a Lipschitz gradient. This allows to directly apply (Eberle et al., 2017)(Corollary 2.6.) which implies that

$$W_2(\pi_t, \pi_\infty) \leq C \exp(-tc),$$

where $c$ is a positive constant and $C$ only depends on $\pi_\infty$ and the initial distribution $\pi_0$. Moreover, the constant $c$ is given explicitly in (Eberle et al., 2017, Theorem 2.3) and is of order $0(e^{-q})$ where $q$ is the dimension of the latent space $\mathcal{Z}$.

We now consider an optimal coupling $\Pi_t$ between $\pi_t$ and $\pi_0$. Given joints samples $((z_t, v_t), (z, v))$ from $\Pi_t$, we consider the following samples in input space $(x_t, x) := (G(z_t), G(z))$. Since $z_t$ and $z$ have marginals $\pi_t$ and $\pi_\infty$, it is easy to see that $x_t \sim \mathbb{P}_t$ and $x \sim \mathbb{Q}$. Therefore, by definition of the $W_2$ distance, we have the following bound:

$$
\begin{aligned}
W_2^2(\mathbb{P}_t, \mathbb{Q}) &\leq \mathbb{E}\big[\|x_t - x\|^2\big] \\
&\leq \int \|G(z_t) - G(z)\|^2 d\Pi_t(z_t, z) \\
&\leq L^2 \int \|z_t - z\|^2 d\Pi_t(z_t, z) \\
&\leq L^2 W_2^2(\pi_t, \pi_\infty) \leq C^2 L^2 \exp(-2tc).
\end{aligned}
$$

The second line uses the definition of $(x_t, x)$ as joint samples obtained by mapping $(z_t, z)$. The third line uses the assumption that $B$ is $L$-Lipschitz. Finally, the last line uses that $\Pi_t$ is an optimal coupling between $\pi_t$ and $\pi_\infty$. $\qquad\square$

**Lemma 13.** *Under Assumption 1, there exists $A > 0$ and $\lambda \in (0, \frac{1}{4}]$ such that*

$$\frac{1}{2} z^{\top t} \nabla U(z) \geq \lambda\left(U(z) + \frac{\gamma^2}{4u}\|z\|^2\right) - A, \qquad \forall z \in \mathcal{Z}, \tag{25}$$

*where $\gamma$ and $u$ are the coefficients appearing in (14). Moreover, $U$ is bounded bellow and has a Lipschitz gradient.*

*Proof.* For simplicity, let's call $u(z) = -\log \eta(z)$, $w(z) = E^\star \circ B_{\theta^\star}(z)$, and denote by $M$ an upper-bound on the Lipschitz constant of $w$ and $\nabla w$ which is guaranteed to be finite by assumption. Hence $U(z) = u(z) + w(z)$. Equation (25) is equivalent to having

$$z^\top \nabla u(z) - 2\lambda u(z) - \frac{\gamma^2}{2u}\|z\|^2 \geq 2\lambda w(z) - z^\top \nabla w(z) - 2A. \tag{26}$$

Using that $w$ is Lipschitz, we have that $w(z) \leq w(0) + M\|z\|$ and $-z^\top \nabla w(z) \leq M\|z\|$. Hence, $2\lambda w(z) - z^\top \nabla w(z) - 2A \leq 2\lambda w(0) + (2\lambda + 1)M\|z\| - 2A$. Therefore, a sufficient condition for (26) to hold is

$$z^\top \nabla u(z) - 2\lambda u(z) - \frac{\gamma^2}{2u}\|z\|^2 \geq +(2\lambda + 1)M\|z\| - 2A + 2\lambda w(0). \tag{27}$$

We will now rely on the strong convexity of $u$, which holds by assumption, and implies the existence of a positive constant $m > 0$ such that

$$-u(z) \geq -u(0) - z^\top \nabla u(z) + \frac{m}{2}\|z\|^2,$$

$$z^\top \nabla u(z) \geq -\|z\|\|\nabla u(0)\| + m\|z\|^2.$$

This allows to write the following inequality,

$$z^\top \nabla u(z) - 2\lambda u(z) - \frac{\gamma^2}{2u} \geq (1-2\lambda)z^\top \nabla u(z) + \lambda(m+\frac{\gamma^2}{2u})\|z\|^2 - 2\lambda u(0)$$
$$\geq (1-\lambda(m+\frac{\gamma^2}{2u}))\|z\|^2 - (1-2\lambda)\|z\|\|\nabla u(0)\| - 2\lambda u(0).$$

Combining the previous inequality with (27) and denoting $M' = \|\nabla u(0)\|$, it is sufficient to find $A$ and $\lambda$ satisfying

$$\left(1 - \lambda\left(m + \frac{\gamma^2}{2u}\right)\right)\|z\|^2 - (M + M' + 2\lambda(M - M'))\|z\| - 2\lambda(u(0) + w(0)) + 2A \geq 0.$$

The l.h.s. in the above equation is a quadratic function in $\|z\|$ and admits a global minimum when $\lambda < \left(m + \frac{\gamma^2}{2u}\right)^{-1}$. The global minimum is always positive provided that $A$ is large enough.

To see that $U$ is bounded below, it suffice to note, by Lipschitzness of $w$, that $w(z) \geq w(0) - M\|z\|$ and by strong convexity of $u$ that

$$u(z) \geq u(0) + M'\|z\| + \frac{m}{2}\|z\|^2.$$

Hence, $U$ is lower-bounded by a quadratic function in $\|z\|$ with positive leading coefficient $\frac{m}{2}$, hence it must be lower-bounded by a constant. Finally, by assumption, $u$ and $w$ have Lipschitz gradients, which directly implies that $U$ has a Lipschitz gradient. □

*Proof of Proposition 3.* By assumption $KL(\mathbb{P}||\mathbb{G}) < +\infty$, this implies that $\mathbb{P}$ admits a density w.r.t. $\mathbb{G}$ which we call $r(x)$. As a result $\mathbb{P}$ admits also a density w.r.t. $\mathbb{Q}$ given by:

$$Z\exp(E^\star(x))r(x).$$

We can then compute the $KL(\mathbb{P}||\mathbb{Q})$ explicitly:

$$KL(\mathbb{P}||\mathbb{Q}) = \mathbb{E}_\mathbb{P}[E] + \log(Z) + \mathbb{E}_\mathbb{P}[\log(r)]$$
$$= -\mathcal{L}_{\mathbb{P},\mathbb{G}}(E^\star) + KL(\mathbb{P}||\mathbb{G}).$$

Since $0$ belongs to $\mathcal{E}$ and by optimality of $E^\star$, we know that $\mathcal{L}_{\mathbb{P},\mathbb{G}}(E^\star) \geq \mathcal{L}_{\mathbb{P},\mathbb{G}}(0) = 0$. The result then follows directly. □

## C.2 TOPOLOGICAL AND SMOOTHNESS PROPERTIES OF KALE

**Topological properties of KALE.** Denseness and smoothness of the energy class $\mathcal{E}$ are the key to guarantee that KALE is a reliable criterion for measuring convergence. We thus make the following assumptions on $\mathcal{E}$:

(A) For all $E \in \mathcal{E}$, $-E \in \mathcal{E}$ and there is $C_E > 0$ such that $cE \in \mathcal{E}$ for $0 \leq c \leq C_E$. For any continuous function $g$, any compact support $K$ in $\mathcal{X}$ and any precision $\epsilon > 0$, there exists a finite linear combination of energies $G = \sum_{i=1}^r a_i E_i$ such that $\sup_{x \in K} |f(x) - G(x)| \leq \epsilon$.

(B) All energies $E$ in $\mathcal{E}$ are Lipschitz in their input with the same Lipschitz constant $L > 0$.

Assumption (A) holds in particular when $\mathcal{E}$ contains feedforward networks with a given number of parameters. In fact networks with a single neuron are enough, as shown in (Zhang et al., 2017, Theorem 2.3). Assumption (B) holds when additional regularization of the energy is enforced during training by methods such as **spectral normalization** Miyato et al. (2018) or **gradient penalty** Gulrajani et al. (2017) as done in Section 6. Proposition 4 states the topological properties of KALE ensuring that it can be used as a criterion for weak convergence. A proof is given in Appendix C.2.1 and is a consequence of (Zhang et al., 2017, Theorem B.1).

**Proposition 14.** *Under Assumptions (A) and (B) it holds that:*

1. *KALE$(\mathbb{P}||\mathbb{G}) \geq 0$ with KALE$(\mathbb{P}||\mathbb{G}) = 0$ if and only if $\mathbb{P} = \mathbb{G}$.*

2. *KALE$(\mathbb{P}||\mathbb{G}^n) \to 0$ if and only if $\mathbb{G}^n \to \mathbb{P}$ under the weak topology.*

### C.2.1 TOPOLOGICAL PROPERTIES OF KALE

In this section we prove Proposition 4. We first start by recalling the required assumptions and make them more precise:

**Assumption 2.** *Assume the following holds:*

- *The set $\mathcal{X}$ is compact.*

- *For all $E \in \mathcal{E}$, $-E \in \mathcal{E}$ and there is $C_E > 0$ such that $cE \in \mathcal{E}$ for $0 \le c \le C_E$. For any continuous function $g$, any compact support $K$ in $\mathcal{X}$ and any precision $\epsilon > 0$, there exists a finite linear combination of energies $G = \sum_{i=1}^{r} a_i E_i$ such that $|f(x) - G(x)| \le \epsilon$ on $K$.*

- *All energies $E$ in $\mathcal{E}$ are Lipschitz in their input with the same Lipschitz constant $L > 0$.*

For simplicity we consider the set $\mathcal{H} = \mathcal{E} + \mathbb{R}$, i.e.: $\mathcal{H}$ is the set of functions $h$ of the form $h = E + c$ where $E \in \mathcal{E}$ and $c \in \mathbb{R}$. In all what follows $\mathcal{P}_1$ is the set of probability distributions with finite first order moments. We consider the notion of weak convergence on $\mathcal{P}_1$ as defined in (Villani, 2009, Definition 6.8) which is equivalent to convergence in the Wasserstein-1 distance $W_1$.

*Proof of Proposition 4 .* We proceed by proving the **separation** properties ($1^{st}$ statement), then the **metrization of the weak topology** ($2^{nd}$ statement).

**Separation.** We have by Assumption 2 that $0 \in \mathcal{E}$, hence by definition $\text{KALE}(PP||\mathbb{G}) \ge \mathcal{F}_{\mathbb{P},\mathbb{G}}(0) = 0$. On the other hand, whenever $\mathbb{P} = \mathbb{G}$, it holds that:

$$\mathcal{F}_{\mathbb{P},\mathbb{G}}(h) = -\int (\exp(-h) + h - 1) d\mathbb{P}, \qquad \forall h \in \mathcal{H}.$$

Moreover, by convexity of the exponential, we know that $\exp(-x) + x - 1 \ge 0$ for all $x \in \mathbb{R}$. Hence, $\mathcal{F}_{\mathbb{P},\mathbb{G}}(h) \le \mathcal{F}_{\mathbb{P},\mathbb{G}}(0) = 0$ for all $h \in \mathcal{H}$. This directly implies that $\text{KALE}(\mathbb{P}|\mathbb{G}) = 0$. For the converse, we will use the same argument as in the proof of (Zhang et al., 2017, Theorem B.1). Assume that $\text{KALE}(\mathbb{P}|\mathbb{G}) = 0$ and let $h$ be in $\mathcal{H}$. By Assumption 2, there exists $C_h > 0$ such that $ch \in \mathcal{H}$ and we have:

$$\mathcal{F}(ch) \le \text{KALE}(\mathbb{P}||\mathbb{G}) = 0.$$

Now dividing by $c$ and taking the limit to 0, it is easy to see that $-\int h d\mathbb{P} + \int h d\mathbb{G} \le 0$. Again, by Assumption 2, we also know that $-h \in \mathcal{H}$, hence, $\int h d\mathbb{P} - \int h d\mathbb{G} \le 0$. This necessarily implies that $\int h d\mathbb{P} - \int h d\mathbb{G} = 0$ for all $h \in \mathcal{H}$. By the density of $\mathcal{H}$ in the set continuous functions on compact sets, we can conclude that the equality holds for any continuous and bounded function, which in turn implies that $\mathbb{P} = \mathbb{G}$.

**Metrization of the weak topology.** We first show that for any $\mathbb{P}$ and $\mathbb{G}$ with finite first moment, it holds that $\text{KALE}(\mathbb{P}|\mathbb{G}) \le LW_1(\mathbb{P},\mathbb{G})$, where $W_1(\mathbb{P},\mathbb{G})$ is the Wasserstein-1 distance between $\mathbb{P}$ and $\mathbb{G}$. For any $h \in \mathcal{H}$ the following holds:

$$\mathcal{F}(h) = -\int h d\mathbb{P} - \int \exp(-h) d\mathbb{G} + 1$$

$$= \int h(x) d\mathbb{G}(x) - h(x') d\mathbb{P}(x')$$

$$- \int \underbrace{(\exp(-h) + h - 1)}_{\ge 0} d\mathbb{G}$$

$$\le \int h(x) d\mathbb{G}(x) - h(x') d\mathbb{P}(x') \le LW_1(\mathbb{P},\mathbb{G})$$

The first inequality results from the convexity of the exponential while the last one is a consequence of $h$ being $L$-Lipschitz. This allows to conclude that $\text{KALE}(\mathbb{P}||\mathbb{G}) \le LW_1(\mathbb{P},\mathbb{G})$ after taking the supremum over all $h \in \mathcal{H}$. Moreover, since $W_1$ metrizes the weak convergence on $\mathcal{P}_1$ (Villani, 2009, Theorem 6.9), it holds that whenever a sequence $\mathbb{G}^n$ converges weakly towards $\mathbb{P}$ in $\mathcal{P}_1$ we also have $W_1(\mathbb{P},\mathbb{G}^n) \to 0$ and thus $\text{KALE}(\mathbb{P}||\mathbb{G}^n) \to 0$. The converse is a direct consequence of (Liu et al., 2017, Theorem 10) since by assumption $\mathcal{X}$ is compact.

**Well-defined learning.**    Assume that for any $\epsilon > 0$ and any $h$ and $h'$ in $\mathcal{E}$ there exists $f$ in $2\mathcal{E}$ such that $\|h + h' - f\|_\infty \leq \epsilon$ then there exists a constant $C$ such that:

$$\mathrm{KALE}(\mathbb{P},\mathbb{Q}) \leq C\mathrm{KALE}(\mathbb{P},\mathbb{G})$$

This means that the proposed learning procedure which first finds the optimal energy $E^\star$ given a base $\mathbb{G}$ by maximum likelihood then minimizes $\mathrm{KALE}(\mathbb{P},\mathbb{G})$ ensures ends up minimizing the distance between the data end the generalized energy-based model $\mathbb{Q}$.

$$\mathrm{KALE}(\mathbb{P},\mathbb{Q}) = \sup_{h \in \mathcal{E}} \mathcal{L}_{\mathbb{P},\mathbb{Q}_\mathbb{G}}(h)$$
$$= -\mathrm{KALE}(\mathbb{P},\mathbb{G}) + \sup_{h \in \mathcal{E}} \mathcal{L}_{\mathbb{P},\mathbb{G}}(h + E^\star)$$

Let's choose $\epsilon = KALE(\mathbb{P},\mathbb{G})$ and let $h \in 2\mathcal{E}$ such that $\|h + E^\star - f\|_\infty \leq \epsilon$. We have by concavity of the function $(\alpha,\beta) \mapsto \mathcal{L}_{\mathbb{P},\mathbb{G}}(\alpha(h + E^\star - f) + \beta f)$ we have that:

$$\mathcal{L}_{\mathbb{P},\mathbb{G}}(h + E^\star) \leq 2\mathcal{L}_{\mathbb{P},\mathbb{G}}(\frac{1}{2}f) - \mathcal{L}_{\mathbb{P},\mathbb{G}}(h + E^\star - f)$$

By assumption, we have that $\|h + E^\star - f\|_\infty \leq \epsilon$, thus $|\mathcal{L}_{\mathbb{P},\mathbb{G}}(h + E^\star - f)| \leq 2\epsilon$. Moreover, we have that $\mathcal{L}_{\mathbb{P},\mathbb{G}}(\frac{1}{2}f) \leq KALE(\mathbb{P},\mathbb{G})$ since $\frac{1}{2}f \in \mathcal{E}$. This ensures that:

$$\mathcal{L}_{\mathbb{P},\mathbb{G}}(h + E^\star) \leq 3\mathrm{KALE}(\mathbb{P},\mathbb{G}).$$

Finally, we have shown that:

$$\mathrm{KALE}(\mathbb{P},\mathbb{Q}) \leq 2\mathrm{KALE}(\mathbb{P},\mathbb{G}).$$

Hence, minimizing $\mathrm{KALE}(\mathbb{P},\mathbb{G})$ directly minimizes $\mathrm{KALE}(\mathbb{P},\mathbb{Q})$.

$\square$

### C.2.2    SMOOTHNESS PROPERTIES OF KALE

We will now prove Theorem 5. We begin by stating the assumptions that will be used in this section:

**(I)** $\mathcal{E}$ is parametrized by a compact set of parameters $\Psi$.

**(II)** Functions in $\mathcal{E}$ are jointly continuous w.r.t. $(\psi, x)$ and are $L$-lipschitz and $L$-smooth w.r.t. the input $x$:

$$\|E_\psi(x) - E_\psi(x')\| \leq L_e\|x - x'\|,$$
$$\|\nabla_x E_\psi(x) - \nabla_x E_\psi(x')\| \leq L_e\|x - x'\|.$$

**(III)** $(\theta, z) \mapsto G_\theta(z)$ is jointly continuous in $\theta$ and $z$, with $z \mapsto G_\theta(z)$ uniformly Lipschitz w.r.t. $z$:

$$\|G_\theta(z) - G_\theta(z')\| \leq L_b\|z - z'\|, \qquad \forall z, z' \in \mathcal{Z}, \theta \in \Theta.$$

There exists non-negative functions $a$ and $b$ defined from $\mathcal{Z}$ to $\mathbb{R}$ such that $\theta \mapsto G_\theta(z)$ are $a$-Lipschitz and $b$-smooth in the following sense:

$$\|G_\theta(z) - G_{\theta'}(z)\| \leq a(z)\|\theta - \theta'\|,$$
$$\|\nabla_\theta G_\theta(z) - \nabla_\theta G_{\theta'}(z)\| \leq b(z)\|\theta - \theta'\|.$$

Moreover, $a$ and $b$ are integrable in the following sense:

$$\int a(z)^2 \exp(2L_e L_b\|z\|)d\eta(z) < \infty, \qquad \int \exp(L_e L_b\|z\|)d\eta(z) < \infty,$$

$$\int b(z)\exp(L_e L_b\|z\|)d\eta(z) < \infty.$$

To simplify notation, we will denote by $\mathcal{L}_\theta(f)$ the expected $\mathbb{G}_\theta$ log-likelihood under $\mathbb{P}$. In other words,

$$\mathcal{L}_\theta(E) := \mathcal{L}_{\mathbb{P},\mathbb{G}_\theta}(E) = -\int E d\mathbb{P} - \log \int \exp(-E) d\mathbb{G}_\theta.$$

We also denote by $p_{E,\theta}$ the density of the model w.r.t. $\mathbb{G}_\theta$,

$$p_{E,\theta} = \frac{\exp(-E)}{Z_{\mathbb{G}_\theta,E}}, \qquad Z_{\mathbb{G}_\theta,E} = \int \exp(-E) d\mathbb{G}_\theta.$$

We write $\mathcal{K}(\theta) := \text{KALE}(\mathbb{P}||\mathbb{G}_\theta)$ to emphasize the dependence on $\theta$.

*Proof of Theorem 5.* To show that sub-gradient methods converge to local optima, we only need to show that $\mathcal{K}$ is Lipschitz continuous and weakly convex. This directly implies convergence to local optima for sub-gradient methods, according to Davis and Drusvyatskiy (2018); Thekumparampil et al. (2019). Lipschitz continuity ensures that $\mathcal{K}$ is differentiable for almost all $\theta \in \Theta$, and weak convexity simply means that there exits some positive constant $C \geq 0$ such that $\theta \mapsto \mathcal{K}(\theta) + C\|\theta\|^2$ is convex. We now proceed to show these two properties.

We will first prove that $\theta \mapsto \mathcal{K}(\theta)$ is weakly convex in $\theta$. By Lemma 15, we know that for any $E \in \mathcal{E}$, the function $\theta \mapsto \mathcal{L}_\theta(E)$ is $M$-smooth for the same positive constant $M$. This directly implies that it is also weakly convex and the following inequality holds:

$$\mathcal{L}_{\theta_t}(E) \leq t\mathcal{L}_\theta(E) + (1-t)\mathcal{L}_{\theta'}(E) + \frac{M}{2}t(1-t)\|\theta - \theta'\|^2.$$

Taking the supremum w.r.t. $E$, it follows that

$$\mathcal{K}(\theta_t) \leq t\mathcal{K}(\theta) + (1-t)\mathcal{K}(\theta') + \frac{M}{2}t(1-t)\|\theta - \theta'\|^2.$$

This means precisely that $\mathcal{K}$ is weakly convex in $\theta$.

To prove that $\mathcal{K}$ is Lipschitz, we will also use Lemma 15, which states that $\mathcal{L}_\theta(E)$ is Lipschitz in $\theta$ uniformly on $\mathcal{E}$. Hence, the following holds:

$$\mathcal{L}_\theta(E) \leq \mathcal{L}_\theta(E) + LC\|\theta - \theta'\|.$$

Again, taking the supremum over $E$, it follows directly that

$$\mathcal{K}(\theta) \leq \mathcal{K}(\theta') + LC\|\theta - \theta'\|.$$

We conclude that $\mathcal{K}$ is Lipschitz by exchanging the roles of $\theta$ and $\theta'$ to get the other side of the inequality. Hence, by the Rademacher theorem, $\mathcal{K}$ is differentiable for almost all $\theta$.

We will now provide an expression for the gradient of $\mathcal{K}$. By Lemma 16 we know that $\psi \mapsto \mathcal{L}_\theta(E_\psi)$ is continuous and by Assumption (I) $\Psi$ is compact. Therefore, the supremum $\sup_{E \in \mathcal{E}} \mathcal{L}_\theta(E)$ is achieved for some function $E_\theta^\star$. Moreover, we know by Lemma 15 that $\mathcal{L}_\theta(E)$ is smooth uniformly on $\mathcal{E}$, therefore the family $(\partial_\theta \mathcal{L}_\theta(E))_{E \in \mathcal{E}}$ is equi-differentiable. We are in position to apply Milgrom and Segal (2002)(Theorem 3) which ensures that $\mathcal{K}(\theta)$ admits left and right partial derivatives given by

$$\begin{aligned}
\partial_e^+ \mathcal{K}(\theta) &= \lim_{\substack{t>0 \\ t \to 0}} \partial_\theta \mathcal{L}_\theta(E_{\theta+te}^\star)^\top e, \\
\partial_e^- \mathcal{K}(\theta) &= \lim_{\substack{t<0 \\ t \to 0}} \partial_\theta \mathcal{L}_\theta(E_{\theta+te}^\star)^\top e,
\end{aligned} \tag{28}$$

where $e$ is a given direction in $\mathbb{R}^r$. Moreover, the theorem also states that $\mathcal{K}(\theta)$ is differentiable iff $t \mapsto E_{\theta+te}^\star$ is continuous at $t = 0$. Now, recalling that $\mathcal{K}(\theta)$ is actually differentiable for almost all $\theta$, it must hold that $E_{\theta+te}^\star \to_{t \to 0} E_\theta^\star$ and $\partial_e^+ \mathcal{K}(\theta) = \partial_e^- \mathcal{K}(\theta)$ for almost all $\theta$. This implies that the two limits in (28) are actually equal to $\partial_\theta \mathcal{L}_\theta(E_\theta^\star)^\top e$. The gradient of $\mathcal{K}$, whenever defined, in therefore given by

$$\nabla_\theta \mathcal{K}(\theta) = Z_{\mathbb{G}_\theta,E_\theta^\star}^{-1} \int \nabla_x E_\theta^\star(G_\theta(z)) \nabla_\theta G_\theta(z) \exp(-E_\theta^\star(G_\theta(z))) \eta(z) \mathrm{d}z.$$

$\square$

**Lemma 15.** *Under Assumptions* ([I]) *to* ([III]), *the functional* $\mathcal{L}_\theta(E)$ *is Lipschitz and smooth in* $\theta$ *uniformly on* $\mathcal{E}$:

$$|\mathcal{L}_\theta(E) - \mathcal{L}_{\theta'}(E)| \leq LC\|\theta - \theta'\|,$$
$$\|\partial_\theta \mathcal{L}_\theta(E) - \partial_\theta \mathcal{L}_{\theta'}(E))\| \leq 2CL(1+L)\|\theta - \theta'\|.$$

*Proof.* By Lemma 16, we have that $\mathcal{L}_\theta(E)$ is differentiable, and that

$$\partial_\theta \mathcal{L}_\theta(E) := \int (\nabla_x E \circ G_\theta) \nabla_\theta G_\theta (p_{E,\theta} \circ G_\theta) d\eta.$$

Lemma 16 ensures that $\|\partial_\theta \mathcal{L}_\theta(E)\|$ is bounded by some positive constant $C$ that is independent from $E$ and $\theta$. This implies in particular that $\mathcal{L}_\theta(E)$ is Lipschitz with a constant $C$. We will now show that it is also smooth. For this, we need to control the difference

$$D := \|\partial_\theta \mathcal{L}_\theta(E) - \partial_\theta \mathcal{L}_{\theta'}(E)\|.$$

We have by triangular inequality:

$$D \leq \underbrace{\int \|\nabla_x E \circ G_\theta - \nabla_x E \circ G_{\theta'}\| \|\nabla_\theta G_\theta\| (p_{E,\theta} \circ G_\theta) d\eta}_{I}$$

$$+ \underbrace{\int \|\nabla_x E \circ G_\theta\| \|\nabla_\theta G_\theta - \nabla_\theta G_{\theta'}\| (p_{E,\theta} \circ G_\theta) d\eta}_{II}$$

$$+ \underbrace{\int \|\nabla_x E \circ G_\theta \nabla_\theta G_\theta\| |p_{E,\theta} \circ G_\theta - p_{E,\theta'} \circ G_{\theta'}| d\eta}_{III}.$$

The first term can be upper-bounded using $L_e$-smoothness of $E$ and the fact that $G_\theta$ is Lipschitz in $\theta$:

$$I \leq L_e \|\theta - \theta'\| \int |a|^2 (p_{E,\theta} \circ G_\theta) d\eta$$
$$\leq L_e C \|\theta - \theta'\|.$$

The last inequality was obtained by Lemma 17. Similarly, using that $\nabla_\theta G_\theta$ is Lipschitz, it follows by Lemma 17 that

$$II \leq L_e \|\theta - \theta'\| \int |b| (p_{E,\theta} \circ G_\theta) d\eta$$
$$\leq L_e C \|\theta - \theta'\|.$$

Finally, for the last term $III$, we first consider a path $\theta_t = t\theta + (1-t)\theta'$ for $t \in [0,1]$, and introduce the function $s(t) := p_{E,\theta_t} \circ G_{\theta_t}$. We will now control the difference $p_{E,\theta} \circ G_\theta - p_{E,\theta'} \circ G_{\theta'}$, also equal to $s(1) - s(0)$. Using the fact that $s_t$ is absolutely continuous we have that $s(1) - s(0) = \int_0^1 s'(t) dt$. The derivative $s'(t)$ is simply given by $s'(t) = (\theta - \theta')^\top (M_t - \bar{M}_t) s(t)$ where $M_t = (\nabla_x E \circ B_{\theta_t}) \nabla_\theta G_{\theta_t}$ and $\bar{M}_t = \int M_t p_{E,\theta_t} \circ G_{\theta_t} d\eta$. Hence,

$$s(1) - s(0) = (\theta - \theta')^\top \int_0^1 (M_t - \bar{M}_t) s(t) dt.$$

We also know that $M_t$ is upper-bounded by $La(z)$, which implies

$$III \leq L_e^2 \|\theta - \theta'\| \int_0^1 \left( \int |a(z)|^2 s(t)(z) d\eta(z) + \left( \int a(z) s(t)(z) d\eta(z) \right)^2 \right)$$
$$\leq L_e^2 (C + C^2) \|\theta - \theta'\|,$$

where the last inequality is obtained using Lemma 17. This allows us to conclude that $\mathcal{L}_\theta(E)$ is smooth for any $E \in \mathcal{E}$ and $\theta \in \Theta$. $\square$

**Lemma 16.** *Under Assumptions (II) and (III), it holds that $\psi \mapsto \mathcal{L}_\theta(E_\psi)$ is continuous, and that $\theta \mapsto \mathcal{L}_\theta(E_\psi)$ is differentiable in $\theta$ with gradient given by*

$$\partial_\theta \mathcal{L}_\theta(E) := \int (\nabla_x E \circ G_\theta) \nabla_\theta G_\theta(p_{E,\theta} \circ G_\theta) d\eta.$$

*Moreover, the gradient is bounded uniformly in $\theta$ and $E$:*

$$\|\nabla_\theta \mathcal{L}_\theta(E)\| \leq L_e \left( \int \exp(-L_e L_b \|z\|) d\eta(z) \right)^{-1} \int a(z) \exp(L_e L_b \|z\|) d\eta(z).$$

*Proof.* To show that $\psi \mapsto \mathcal{L}_\theta(E_\psi)$ is continuous, we will use the dominated convergence theorem. We fix $\psi_0$ in the interior of $\Psi$ and consider a compact neighborhood $W$ of $\psi_0$. By assumption, we have that $(\psi, x) \mapsto E_\psi(x)$ and $(\psi, z) \mapsto E_\psi(G_\theta(z))$ are jointly continuous. Hence, $|E_\psi(0)|$ and $|E_\psi(G_\theta(0))|$ are bounded on $W$ by some constant $C$. Moreover, by Lipschitz continuity of $x \mapsto E_\psi$, we have

$$|E_\psi(x)| \leq |E_\psi(0)| + L_e \|x\| \leq C + L_e \|x\|,$$
$$\exp(-E(G_\theta(z))) \leq \exp(-E(G_\theta(0))) \exp(L_e L_b \|z\|) \leq \exp(C) \exp(L_e L_b \|z\|).$$

Recalling that $\mathbb{P}$ admits a first order moment and that by Assumption (III), $\exp(L_e L_b \|z\|)$ is integrable w.r.t. $\eta$, it follows by the dominated convergence theorem and by composition of continuous functions that $\psi \mapsto \mathcal{L}_\theta(E_\psi)$ is continuous in $\psi_0$.

To show that $\theta \mapsto \mathcal{L}_\theta(E_\psi)$ is differentiable in $\theta$, we will use the differentiation lemma in (Klenke, 2008, Theorem 6.28). We first fix $\theta_0$ in the interior of $\Theta$, and consider a compact neighborhood $V$ of $\theta_0$. Since $\theta \mapsto |E(G_\theta(0))|$ is continuous on the compact neighborhood $V$ it admits a maximum value $C$; hence we have using Assumptions (II) and (III) that

$$\exp(-E(G_\theta(z))) \leq \exp(-E(G_\theta(0))) \exp(L_e L_b \|z\|) \leq \exp(C) \exp(L_e L_b \|z\|).$$

Along with the integrability assumption in Assumption (III), this ensures that $z \mapsto \exp(-E(G_\theta(z)))$ is integrable w.r.t $\eta$ for all $\theta$ in $V$. We also have that $\exp(-E(G_\theta(z)))$ is differentiable, with gradient given by

$$\nabla_\theta \exp(-E(G_\theta(z))) = \nabla_x E(G_\theta(z)) \nabla_\theta G_\theta(z) \exp(-E(G_\theta(z))).$$

Using that $E$ is Lipschitz in its inputs and $G_\theta(z)$ is Lipschitz in $\theta$, and combining with the previous inequality, it follows that

$$\|\nabla_\theta \exp(-E(G_\theta(z)))\| \leq \exp(C) L_e a(z) \exp(L_e L_b \|z\|),$$

where $a(z)$ is the location dependent Lipschitz constant introduced in Assumption (III). The r.h.s. of the above inequality is integrable by Assumption (III) and is independent of $\theta$ on the neighborhood $V$. Thus (Klenke, 2008, Theorem 6.28) applies, and it follows that

$$\nabla_\theta \int \exp(-E(G_{\theta_0}(z))) d\eta(z) = \int \nabla_x E(G_{\theta_0}(z)) \nabla_\theta G_{\theta_0}(z) \exp(-E(G_{\theta_0}(z))) d\eta(z).$$

We can now directly compute the gradient of $\mathcal{L}_\theta(E)$,

$$\nabla_\theta \mathcal{L}_\theta(E) = \left( \int \exp(-E(G_{\theta_0})) d\eta \right)^{-1} \int \nabla_x E(G_{\theta_0}) \nabla_\theta G_{\theta_0} \exp(-E(G_{\theta_0})) d\eta.$$

Since $E$ and $G_\theta$ are Lipschitz in $x$ and $\theta$ respectively, it follows that $\|\nabla_x E(G_{\theta_0}(z))\| \leq L_e$ and $\|\nabla_\theta G_{\theta_0}(z)\| \leq a(z)$. Hence, we have

$$\|\nabla_\theta \mathcal{L}_\theta(E)\| \leq L_e \int a(z)(p_{E,\theta} \circ G_\theta(z)) d\eta(z).$$

Finally, Lemma 17 allows us to conclude that $\|\nabla_\theta \mathcal{L}_\theta(E)\|$ is bounded by a positive constant $C$ independently from $\theta$ and $E$. $\square$

**Lemma 17.** *Under Assumptions (II) and (III), there exists a constant $C$ independent from $\theta$ and $E$ such that*

$$\int a^i(z)(p_{E,\theta} \circ G_\theta(z)) d\eta(z) < C, \tag{29}$$

$$\int b(z)(p_{E,\theta} \circ G_\theta(z)) d\eta(z) < C,$$

*for $i \in 1, 2$.*

| Model | FID |
|---|---|
| **Cifar10 Unsupervised** | |
| PixelCNN Oord et al. (2016) | 65.93 |
| PixelIQN Ostrovski et al. (2018) | 49.46 |
| EBM Radford et al. (2015) | 38.2 |
| WGAN-GP Gulrajani et al. (2017) | 36.4 |
| NCSN Ho and Ermon (2016) | 25.32 |
| SNGAN Miyato et al. (2018) | 21.7 |
| GEBM (ours) | **19.31** |
| **Cifar10 Supervised** | |
| BigGAN Donahue and Simonyan (2019) | 14.73 |
| SAGAN Zenke et al. (2017) | 13.4 |
| **ImageNet Conditional** | |
| PixelCNN | 33.27 |
| PixelIQN | 22.99 |
| EBM | 14.31 |
| **ImageNet Supervised** | |
| SNGAN | 20.50 |
| GEBM (ours) | **13.94** |

Table 4: FID scores on ImageNet and CIFAR-10.

*Proof.* By Lipschitzness of $E$ and $G_\theta$, we have $\exp(-L_e L_b \|z\|) \le \exp(E(G_\theta(0)) - E(G_\theta(z)) \le \exp(L_e L_b \|z\|)$, thus introducing the factor $\exp(E(B_{\theta_0}(0))$ in (29) we get

$$\int a^i(z)(p_{E,\theta} \circ G_\theta(z))d\eta(e) \le L_e \left( \int \exp(-L_e L_b \|z\|) d\eta(z) \right)^{-1} \int a(z)^i \exp(L_e L_b \|z\|) d\eta(z),$$

$$\int b(z)(p_{E,\theta} \circ G_\theta(z))d\eta(z) \le L_e \left( \int \exp(-L_e L_b \|z\|) d\eta(z) \right)^{-1} \int b(z) \exp(L_e L_b \|z\|) d\eta(z).$$

The r.h.s. of both inequalities is independent of $\theta$ and $E$, and finite by the integrability assumptions in Assumption (III). □

## D IMAGE GENERATION

Figures 3 and 4 show sample trajectories using Algorithm 3 with no friction $\gamma = 0$ for the 4 datasets. It is clear that along the same MCMC chain, several image modes are explored. We also notice the transition from a mode to another happens almost at the same time for all chains and corresponds to the gray images. This is unlike Langevin or when the friction coefficient $\gamma$ is large as in Figure 5. In that case each chain remains within the same mode.

Table 4 shows further comparisons with other methods on Cifar10 and ImageNet 32x32.

## E DENSITY ESTIMATION

Figure Figure 7 (left) shows the error in the estimation of the log-partition function using both methods (KALE-DV and KALE-F). KALE-DV estimates the negative log-likelihood on each batch of size 100 and therefore has much more variance than KALE-F which maintains the amortized estimator of the log-partition function.

Figure Figure 7 (right) shows the evolution of the negative log-likelihood (NLL) on both training and test sets per epochs for RedWine and Whitewine datasets. The error decreases steadily in the case of KALE-DV and KALE-F while the error gap between the training and test set remains controlled.

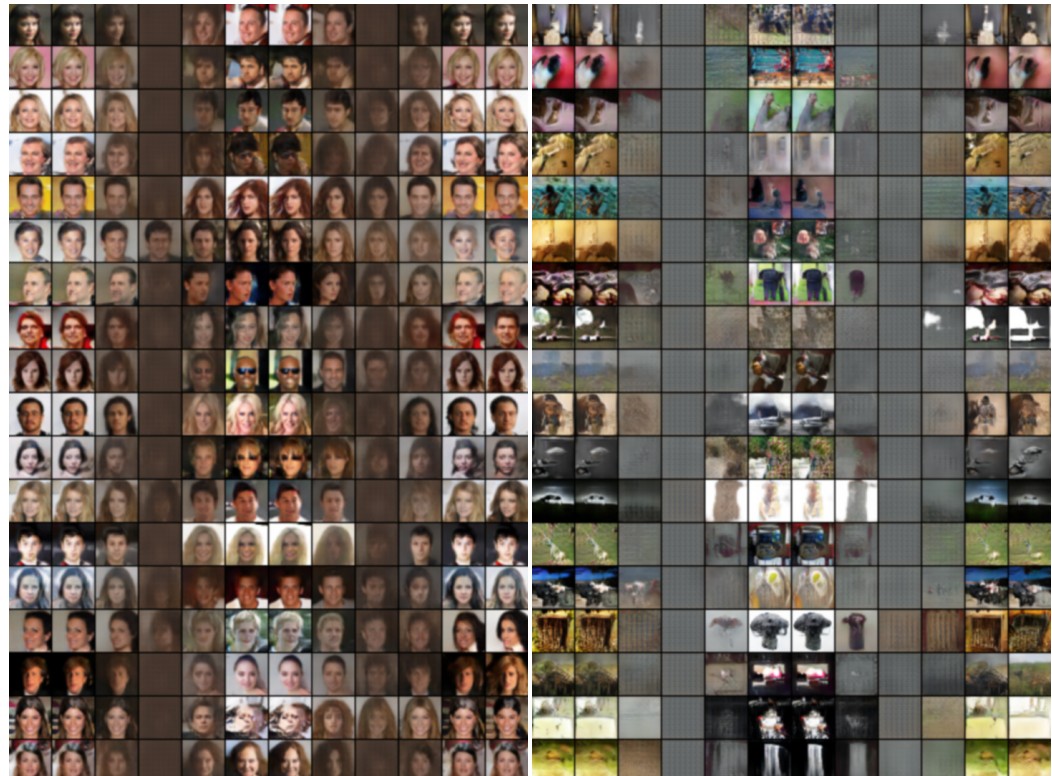

Figure 3: Samples from the GEBM at different stages of sampling using Algorithm 3 and inverse temperature $\beta = 1$, on CelebA (Left), Imagenet (Right). Each row represents a sampling trajectory from early stages (leftmost images) to later stages (rightmost images).

Larger gaps are observed for both direct maximum likelihood estimation and Contrastive divergence although the training NLL tends to decrease faster than for KALE.

## F    ALGORITHMS

**Estimating the variational parameter.** Optimizing (9) exactly over $A$ yields (8), with the optimal $A$ equal to $\tilde{A} = \log(\frac{1}{M}\sum_{m=1}^{M} \exp(-E(Y_m)))$. However, to maintain an amortized estimator of the log-partition we propose to optimize (9) iteratively using second order updates:

$$A_{k+1} = A_k - \lambda(\exp(A_k - \tilde{A}_{k+1}) - 1), \qquad A_0 = \tilde{A}_0 \tag{30}$$

where $\lambda$ is a learning rate and $\tilde{A}_{k+1}$ is the empirical log-partition function estimated from a batch of new samples. By leveraging updates from previous iterations, $A$ can yield much more accurate estimates of the log-partition function as confirmed empirically in Figure 7 of Appendix E.

**Tempered GEBM.** It can be preferable to sample from a *tempered* version of the model by rescaling the energy $E$ by an *inverse temperature* parameter $\beta$, thus effectively sampling from $\mathbb{Q}$. *High temperature* regimes ($\beta \to 0$) recover the base model $\mathbb{G}$ while *low temperature* regimes ($\beta \to \infty$) essentially sample from minima of the energy $E$. As shown in Section 6, low temperatures tend to produce better sample quality for natural image generation tasks.

**Training**    In Algorithm 1, we describe the general algorithm for training a GEBM which alternates between gradient steps on the energy and the generator. An additional regularization, denoted by $I(\psi)$ is used to ensure conditions of Proposition 4 and Theorem 5 hold. $I(\psi)$ can include $L_2$ regularization over the parameters $\psi$, a gradient penalty as in Gulrajani et al. (2017) or Spectral normalization Miyato et al. (2018). The energy can be trained either using the estimator in (8) (KALE-DV) or the one in (9) (KALE-F) depending on the variable $\mathcal{C}$.

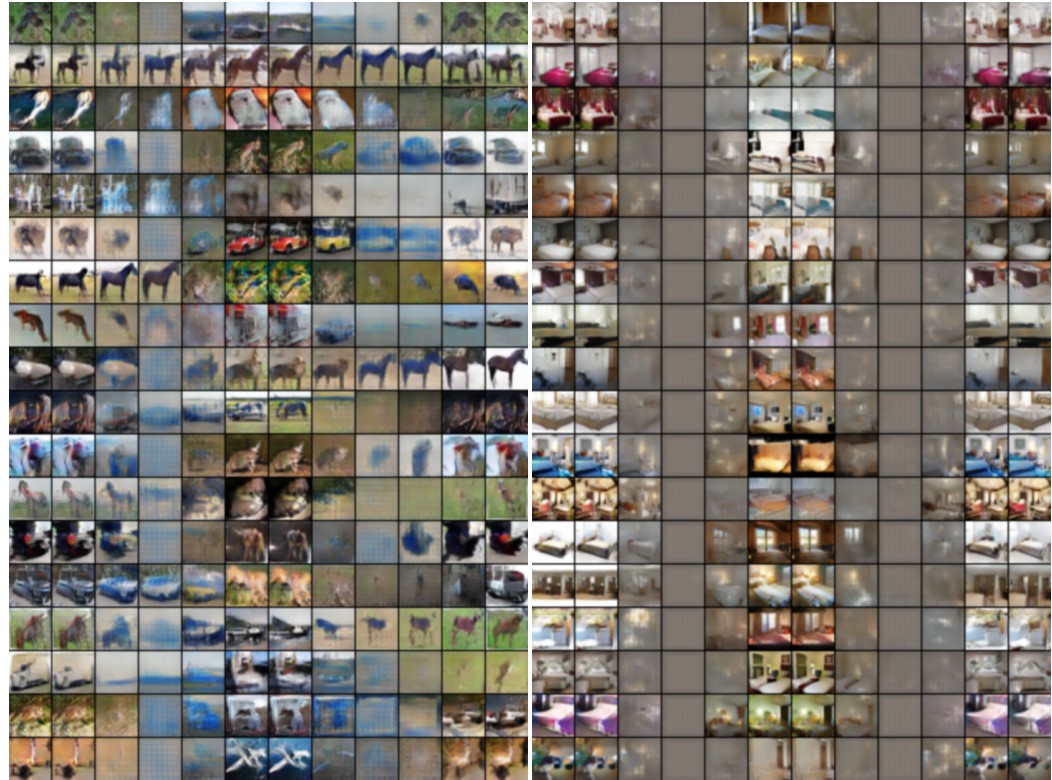

Figure 4: Samples from the GEBM at different stages of sampling using Algorithm 3 and inverse temperature $\beta = 1$, on Cifar10 and LSUN (Right). Each row represents a sampling trajectory from early stages (leftmost images) to later stages (rightmost images).

**Sampling** In Algorithm 3, we describe the MCMC sampler proposed in Sachs et al. (2017) which is a time discretization of (14).

---

**Algorithm 2** Overdamped Langevin Algorithm

---
1: **Input** $\lambda, \gamma, u, \eta, E, G$
2: **Ouput** $X_T$
3: $Z_0 \sim \eta$ // Sample Initial latent from $\eta$.
4: **for** $t = 0, ..., T$ **do**
5: $\quad Y_{t+1} \leftarrow \nabla_z \log \eta(Z_t) - \nabla_z E \circ B(Z_t)$ // Evaluating $\nabla_z \log(\nu(Z_{t+1}))$ using (4).
6: $\quad W_{t+1} \sim \mathcal{N}(0, I)$ // Sample standard Gaussian noise
7: $\quad Z_{t+1} \leftarrow Z_t + \lambda Y_{t+1} + \sqrt{2\lambda} W_{t+1}$
8: **end for**
9: $X_T \leftarrow G(Z_T)$

---

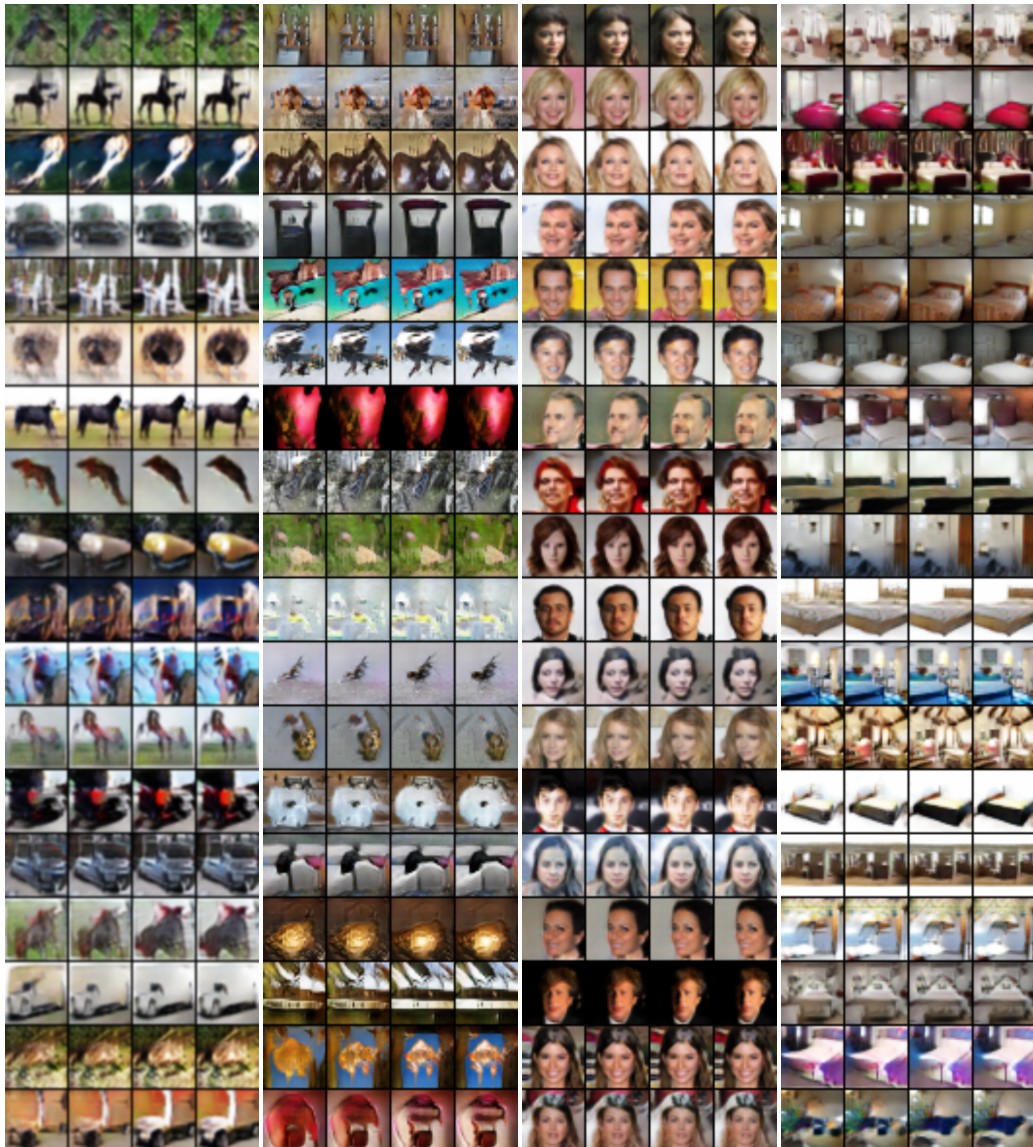

Figure 5: Samples from the tempered GEBM at different stages of sampling using langevin and inverse temperature $\beta = 100$, on Cifar10 (Left), Imagenet (Middle-left), CelebA (Middle-Right) and LSUN (Right). Each row represents a sampling trajectory from early stages (leftmost images) to later stages (rightmost images).

---

**Algorithm 3** Kinetic Langevin Algorithm

---

1: **Input** $\lambda, \gamma, u, \eta, E, G$
2: **Ouput** $X_T$
3: $Z_0 \sim \eta$ // Sample Initial latent from $\eta$.
4: **for** $t = 0, ..., T$ **do**
5: $\quad Z_{t+1} \leftarrow Z_t + \frac{\lambda}{2} V_t$
6: $\quad Y_{t+1} \leftarrow \nabla_z \log \eta(Z_{t+1}) - \nabla_z E \circ B(Z_{t+1})$ // Evaluating $\nabla_z \log(\nu(Z_{t+1}))$ using (4).
7: $\quad V_{t+1} \leftarrow V_t + \frac{u\lambda}{2} Y_{t+1}$.
8: $\quad W_{t+1} \sim \mathcal{N}(0, I)$ // Sample standard Gaussian noise
9: $\quad \tilde{V}_{t+1} \leftarrow \exp(-\gamma\lambda) V_{t+\frac{1}{2}} + \sqrt{u(1 - \exp(-2\gamma\lambda))} W_{t+1}$
10: $\quad V_{t+1} \leftarrow \tilde{V}_{t+1} + \frac{u\lambda}{2} Y_{t+1}$
11: $\quad Z_{t+1} \leftarrow Z_{t+1} + \frac{\lambda}{2} V_{t+1}$
12: **end for**
13: $X_T \leftarrow G(Z_T)$

---

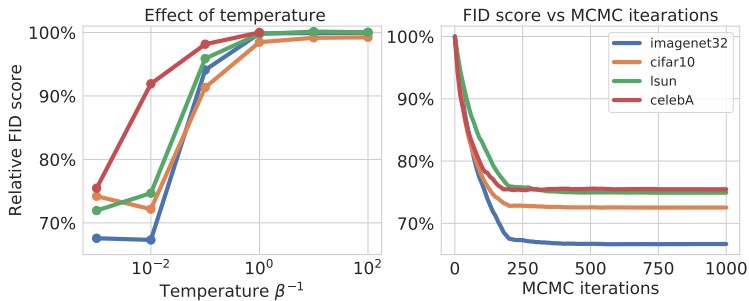

Figure 6: Relative FID score: ratio between FID score of the GEBM $\mathbb{Q}_{\mathbb{G},E}$ and its base $\mathbb{G}$. (Left) Evolution of the ratio for increasing temperature on the 4 datasets after 1000 iterations of (14). (Right) Evolution of the same ratio during MCMC iteration using (14).

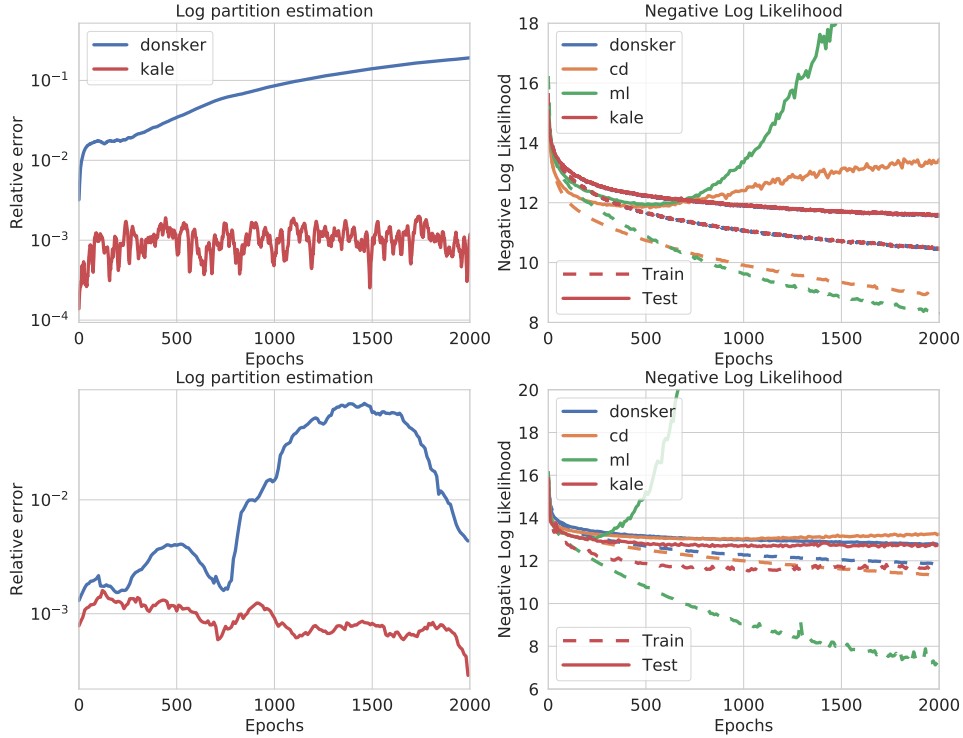

Figure 7: (Left): Relative error $\frac{|\hat{c} - c^\star|}{|\hat{c}| + |c^\star|}$ on the estimation of the ground truth log-partition function $c^\star$ by $\hat{c}$ using either KALE-DV or KALE-F vs training Epochs on RedWine (Top) and WhiteWine (Bottom) datasets. (Right): Negative log likelihood vs training epochs on both training and test set for 4 different learning methods (KALE-DV,KALE-F, CD and ML) on RedWine dataset.

## G   EXPERIMENTAL DETAILS

In all experiments, we use **regularization** which is a combination of $L_2$ norm and a variant of the gradient penalty Gulrajani et al. (2017). For the image generation tasks, we also employ spectral normalization Miyato et al. (2018). This is to ensure that the conditions in Proposition 4 and Theorem 5 hold. We **pre-condition** the gradient as proposed in Simsekli et al. (2020) to stabilize training, and to avoid taking large noisy gradient steps due to the exponential terms in (8) and (9). We also use the second-order updates in (30) for the variational constant $c$ whenever it is learned.

### G.1   IMAGE GENERATION

| $z \in \mathbb{R}^{100} \sim \mathcal{N}(0,I)$ |
|---|
| dense $\rightarrow M_g \times M_g \times 512$ |
| $4 \times 4$, stride$= 2$ deconv. BN 256 ReLU |
| $4 \times 4$, stride$= 2$ deconv. BN 128 ReLU |
| $4 \times 4$, stride$= 2$ deconv. BN 64 ReLU |
| $3 \times 3$, stride$= 1$ conv. 3 Tanh |

Table 5: Base/Generator of SNGAN ConvNet: $M_g = 4$.

| RGB image $x \in \mathbb{R}^{M \times M \times 3}$ |
|---|
| $3 \times 3$, stride$= 1$ conv 64 lReLU |
| $4 \times 4$, stride$= 2$ conv 64 lReLU |
| $3 \times 3$, stride$= 1$ conv 128 lReLU |
| $4 \times 4$, stride$= 2$ conv 128 lReLU |
| $3 \times 3$, stride$= 1$ conv 256 lReLU |
| $4 \times 4$, stride$= 2$ conv 256 lReLU |
| $3 \times 3$, stride$= 1$ conv 512 lReLU |
| dense $\rightarrow 1$. |

Table 6: Energy/Discriminator of SNGAN ConvNet: $M = 32$.

| RGB image $x \in \mathbb{R}^{M \times M \times 3}$ |
|---|
| ResBlock down 128 |
| ResBlock down 128 |
| ResBlock 128 |
| ResBlock 128 |
| ReLu |
| Global sum pooling |
| dense $\rightarrow 1$ |

Table 7: Energy/Discriminator of SNGAN ResNet.

| $z \in \mathbb{R}^{100} \sim \mathcal{N}(0,I)$ |
|---|
| dense, $4 \times 4 \times 256$ |
| ResBlock up 256 |
| ResBlock up 256 |
| ResBlock up 256 |
| BN, ReLu, $3 \times 3$ conv, Tanh |

Table 8: Base/Generator of SNGAN ResNet.

**Network Architecture** Table 5 and Table 6 show the network architectures used for the GEBM in the case of SNGAN ConvNet. Table 5 and Table 6 show the network architectures used for the GEBM in the case of SNGAN ResNet. The residual connections of each residual block consists of two convolutional layers proceeded by a BatchNormalization and ReLU activation: **BN+ReLU+Conv+BN+ReLU+Conv** as in (Miyato et al., 2018, Figure 8).

**Training:** We train both base and energy by alternating 5 gradient steps to learn the energy vs 1 gradient step to learn the base. For the first two gradient iterations and after every 500 gradient iterations on base, we train the energy for 100 gradient steps instead of 5. We then train the model up to 150000 gradient iterations on the base using a batch-size of 128 and Adam optimizer Kingma and Ba (2014) with initial learning rate of $10^{-4}$ and parameters $(0.5,.999)$ for both energy and base.

**Scheduler:** We decrease the learning rate using a scheduler that monitors the FID score in a similar way as in Bińkowski et al. (2018); Arbel et al. (2018). More precisely, every 2000 gradient iterations on the base, we evaluate the FID score on the training set using 50000 generated samples from the base and check if the current score is larger than the score 20000 iterations before. The learning rate is decreased by a factor of 0.8 if the FID score fails to decrease for 3 consecutive times.

**Sampling:** For (DOT) Tanaka (2019), we use the following objective:

$$z \mapsto \|z - z_y + \epsilon\| + \frac{1}{k_{eff}} E \circ G(z) \tag{31}$$

where $z_y$ is sampled from a standard Gaussian, $\epsilon$ is a perturbation meant to stabilize sampling and $k_{eff}$ is the estimated Lipschitz constant of $E \circ B$. Note that (31) uses a flipped sign for the $E \circ B$ compared to Tanaka (2019). This is because $E$ plays the role of $-D$ where $D$ is the discriminator in Tanaka (2019). Introducing the minus sign in (31) leads to a degradation in performance. We perform 1000 gradient iterations with a step-size of 0.0001 which is also decreased by a factor of 10 every 200 iterations as done for the proposed method. As suggested by the authors of Tanaka (2019) we perform the following projection for the gradient before applying it:

$$g \leftarrow g - \frac{(g^\top z)}{\sqrt{q}} z.$$

We set the perturbation $\epsilon$ to $0.001$ and $k_{eff}$ to $1$ which was also shown in Tanaka (2019) to perform well. In fact, we found that estimating the Lipschitz constant by taking the maximum value of $\|\nabla E \circ G(z)\|$ over 1000 latent samples according to $\eta$ lead to higher values for $k_{eff}$: ( Cifar10: 9.4, CelebA : 7.2, ImageNet: 4.9, Lsun: 3.8). However, those higher values did not perform as well as setting $k_{eff} = 1$.

For (IHM) Turner et al. (2019) we simply run the MCMC chain for 1000 iterations.

## G.2 DENSITY ESTIMATION

**Pre-processing** We use code and pre-processing steps from Wenliang et al. (2019) which we describe here for completeness. For RedWine and WhiteWine, we added uniform noise with support equal to the median distances between two adjacent values. That is to avoid instabilities due to the quantization of the datasets. For Hepmass and MiniBoone, we removed ill-conditioned dimensions as also done in Papamakarios et al. (2017). We split all datasets, except HepMass into three splits. The test split consists of $10\%$ of the total data. For the validation set, we use $10\%$ of the remaining data with an upper limit of 1000 to reduce the cost of validation at each iteration. For HepMass, we used the sample splitting as done in Papamakarios et al. (2017). Finally, the data is whitened before fitting and the whitening matrix was computed on at most 10000 data points.

**Regularization:** We set the regularization parameter to $0.1$ and use a combination of $L_2$ norm and a variant of the gradient penalty Gulrajani et al. (2017):

$$I(\psi)^2 = \frac{1}{d_\psi} \|\psi\|^2 + \mathbb{E}\left[\|\nabla_x f_\psi(\widetilde{X})\|^2\right]$$

**Network Architecture.** For both base and energy, we used an NVP Dinh et al. (2016) with 5 NVP layers each consisting of a shifting and scaling layer with two hidden layers of 100 neurons. We do not use Batch-normalization.

**Training:** In all cases we use Adam optimizer with learning rate of $0.001$ and momentum parameters $(0.5, 0.9)$. For both KALE-DV and KALE-F, we used a batch-size of 100 data samples vs 2000 generated samples from the base in order to reduce the variance of the estimation of the energy. We alternate 50 gradient steps on the energy vs 1 step on the base and further perform 50 additional steps on the energy for the first two gradient iterations and after every 500 gradient iterations on base. For Contrastive divergence, each training step is performed by first producing 100 samples from the model using 100 Langevin iterations with a step-size of $10^{-2}$ and starting from a batch of 100 data-samples. The resulting samples are then used to estimate the gradient of the of the loss.

For (CD), we used 100 Langevin iterations for each learning step to sample from the EBM. This translates into an improved performance at the expense of increased computational cost compared to the other methods. All methods are trained for 2000 epochs with batch-size of 100 (1000 on Hepmass and Miniboone datasets) and fixed learning rate $0.001$, which was sufficient for convergence.

## G.3 ILLUSTRATIVE EXAMPLE IN FIGURE 1

We consider parametric functions $G_\theta^{(1)}$ and $G_\theta^{(2)}$ from $\mathbb{R}$ to $\mathbb{R}$ of the form:

$$G_\theta^{(1)}(x) = sin(8\pi Wx)/(1+4\pi Bx), \qquad G_\theta^{(2)}(x) = 4\pi W'x + b$$

with $\theta = (W, B, W', b)$. we also call $\theta^\star = (1, 1, 1, 0)$. In addition, we consider a sigmoid like function $h$ from $[0,1]$ to $[0,1]$ of the form:

$$\widetilde{z} = tan(\pi(z - \frac{1}{2})), \qquad h(z) = \frac{1}{2}\left(z + \frac{1}{1+\exp(-9\widetilde{z})}\right).$$

**Data generation** : To generate a data point $X = (X_1, X_2)$, we consider the following simple generative model:

- Sample a uniform r.v. $Z$ from $[0,1]$.
- Apply the distortion function $h$ to get a latent sample $Y = h(Z)$.

- Generate point $X$ using $X_1 = G_{\theta^\star}^{(1)}(Y)$ and $X_2 = G_{\theta^\star}^{(2)}(Y)$.

Hence, the data are supported on the 1-d line defined by the equation $X_2 = G_{\theta^\star}^{(2)}(X_1)$.

**GAN** For the generator we sample $Z$ uniformly from $[0,\ 1]$ then generate a sample $(X_1, X_2) = (G_\theta^{(1)}(Z), G_\theta^{(2)}(Z))$. The goal is to learn $\theta$.

For the discriminator, we used an MLP with 6 layers and 10 hidden units.

**GEBM** For the base we use the same generator as in the GAN model. For the energy we use the same MLP as discriminator of the GAN model.

**EBM** To ensure tractability of the likelihood, we use the following model:

$$X_2|X_1 \sim \mathcal{N}(G_\theta^{(2)}(X_1), \sigma_0)$$
$$X_1 \sim MoG((\mu_1, \sigma_1), (\mu_2, \sigma_2))$$

$MoG((\mu_1, \sigma_1), (\mu_2, \sigma_2))$ refers to a Mixture of two gaussians with mean and variances $\mu_i$ and $\sigma_i$. We learn each of the parameters $(\theta, \sigma_0, \mu_1, \sigma_1, \mu_2, \sigma_2)$ by maximizing the likelihood.

Both GAN and GEBM have the capacity to recover the the exact support by finding the optimal parameter $\theta^\star$. For the EBM, when $\theta = \theta^\star$, the mean $G_{\theta^\star}(X_1)$ of the conditional gaussian $X_2|X_1$ draws a line which matches the data support exactly, i.e.: $X_2 = G_{\theta^\star}^{(2)}(X_1)$.

### G.4 BASE/GENERATOR COMPLEXITY

To investigate the effect of model complexity of the performance gap between GANs and GEBMs, we performed additional experiments using the setting of Figure 1. Now we allow the generator/base network to better model the hidden transformation $h$ that produces the first coordinate $X_1$ given the latent noise. We choose $G_\theta^{(1)}$ to be either a one hidden layer network or an MLP with 3 hidden layers both with leaky ReLU activation, instead of a simple linear transform as previously done in Appendix G.3. The network has universal approximation capability that depends on the number of units. This provides a direct control over the complexity of the generator/base. We then varied the number of hidden units from 1 to $5*10^4$ units for the one hidden layer network and from 10 to $5*10^3$ units per layer for the MLP. Note that the MLP with $5*10^3$ units per layer stores a matrix of size $2.5*10^7$ and thus contains 2 orders of magnitudes more parameters than the widest shallow network with $5*10^4$ units. We then compared the performance of the GAN and GEBM using the Sinkhorn divergence Feydy et al. (2019) between each model and the data distribution. In all experiments, we used the same discriminator/energy network described in Appendix G.3. Results are provided in Figure 8.

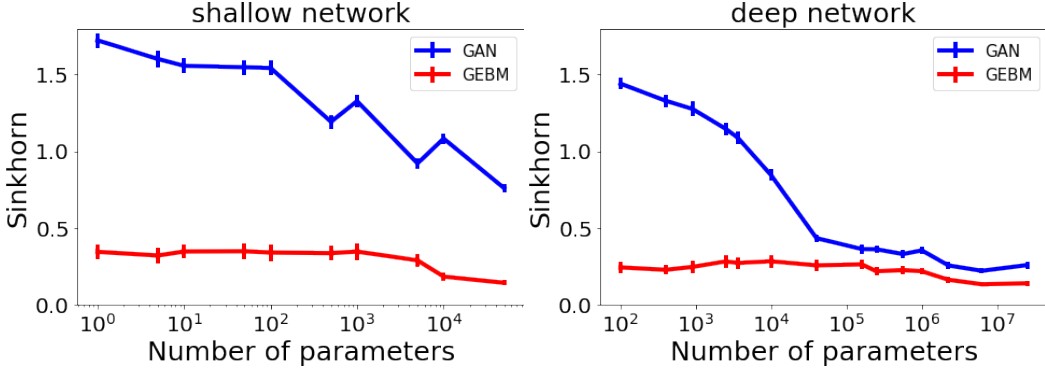

Figure 8: Sinkhorn divergence between data generating distribution and the trained model (either GAN or GEBM) vs number of hidden units in $G_\theta^{(1)}$. The left figure represents the one hidden layer network and right one is for the MLP. Each data point represents the average over 20 independent runs for each choice of number of hidden units.

**Estimating the Sinkhorn divergence.**    The Sinkhorn is computed using $6000$ samples from the data and the model, with squared euclidean distance as a ground cost and using a regularization $\epsilon = 1e-3$. We then repeat the procedure $5$ times and average the result to get the final estimate of the Sinkhorn distance for a given run.

**Training.**    Each run optimizes the parameters of the model using Adam optimizer $(\beta_1 = .5, \beta_2 = .99)$, learning rate $lr = 1e-4$ for the energy/discriminator and $lr = 1e-5$ for the base/generator and weight decay of $1e-2$ for the base/generator. Training is performed using KALE for $2000$ epochs using a batch size of $5000$ and $10$ gradient iterations for the energy/discriminator per base/generator iteration. We use the gradient penalty for the energy/discriminator with a penalty parameter of $0.01$. We then perform early stopping and retain the best performing model on a validation set.

**Observations**    We make the following observations from Figure 8: the GAN generator indeed improves when we increase the number of hidden units. The performance of the GEBM remains stable as the number of hidden units increases. The performance of the GEBM is always better than the GAN, although we can see the GAN converging towards the GEBM. GEBM with a simpler base already outperforms the GAN with more powerful generators. The gap between the GEBM and the GAN reduces as the GAN becomes more expressive. Using a deeper network further reduces the gap compared to a shallow network.

These observations support the prior discussion that the energy witnesses a remaining difference between the generator and training samples, as long as it is not flat. This information allows the GEBM to perform better than a GAN that ignores it. The performance gap between the GEBM and the GAN reduces as the generator becomes more powerful and forces the energy to be more flat. This is consistent with the result in Proposition 3.

