# OpenReview forum: "Generalized Energy Based Models"
_ICLR.cc/2021/Conference — ICLR 2021 Poster_

### Official Review · AnonReviewer2 · 2020-10-23
**Lack of Novelty**

**Rating:** 6
**Confidence:** 3

**Review:**

This paper proposes a framework called GEBM that combine an implicit generator and an EBM to define a probabilistic model on low dimensional manifold. Specifically, the implicit generator defines the base distribution, and the EBM refines the base. Finally, this method is equivalent to define an EBM on the latent space of the implicit generator together with a mapping from the latent space to the data space. The authors propose to use the KALE to train the generator, and provide a theoretical guarantee about the validness of the KALE.

I personally enjoy reading this paper. It provides a good method to solve the problem where the EBM cannot model the distribution whose support is a low-dimensional  manifold in the data space. However, my main concern is about the contribution of this paper, which can be summarized as follows:
1. The novelty. This paper mainly has two contributions: the generalized ebm framework and the KALE objective function. For the first contribution, the difference between the related work is discussed. But what’s the advantages? When the base distribution is an implicit model, GEBM is still a GAN-like model: cannot estimate density. When the base distribution is a flow model, GEBM is still an EBM: cannot provide tractable partition function.
2. The validness of the proposed method. If the base distribution is an implicit model, which means that the support is only a subset of the data space. However, most implicit models, especially GANs, suffer from the mode collapse problem. In this case, the distribution defined by GEBM cannot recover the data distribution. Is there any explanation about the mode collapse problem in GEBM?


Some minor concerns about this paper are provided as follows:
1. The experimental results are chaotic. First, the neural architectures are not introduced, which is the key factor in the performance of GAN-like models. Second, Figure 1 needs more explanation. What is the architectures of the models? Which GAN variant is used? Is EBM trained by MLE using MCMC or score matching?
2. What’s the main purpose of using the flow model as the density estimator? As mentioned in Proposition 2, GEBM is equivalent an EBM. Besides, in the experiments, the energy model is also an NVP. More explanation should be added to improve the clarity. Further, how NVP is trained with CD?

---

> ### Author Response · Authors · 2020-11-13
> **Thank you for your review**
>
> Thank you for your review and for constructive remarks. We are happy to hear that you enjoyed reading the paper and that you found the proposed method as a good alternative to EBMs for data with support in a lower-dimensional manifold. We hope the following discussion answers your questions.
>
> We understand that the first two questions you raised are about the advantages/limitations of the proposed model (GEBM) (Density estimation on Manifolds and Mode collapse) more than they are about the novelty of the model or the validity of the proposed method. Thus we thank you for encouraging us to strengthen the discussion on the advantages/limitations of the model which we will include in the revised version of the paper. We refer to the related work section concerning the novelty and to the theory results for the validity of the method.
> ### 1- The advantages of the method:
> #### 1- Density estimation on Manifold data:
> As you pointed out the GEBM has an advantage over EBM in that it can model data supported on a low-dimensional manifold. Under the manifold assumption, the data do not admit a density on the whole space, as a consequence, it is not possible to estimate this (undefined) quantity.  What GEBM offers is the possibility to estimate a density relative to the support of the generator. Thus if the model learns the low dimensional support of data using the generator, the GEBM provides a way to model the density on that learned manifold.
>  #### 2- Mode collapse:
> This is an important question, thank you for raising it. Mode collapse can happen if the model doesn’t have enough capacity to cover the data support. One setting where this can occur is when the data is supported over multiple disjoint sub-manifolds (e.g. multiple CIFAR classes).
> Multimodality can be a challenge in GANs: the latent noise is usually unimodal, however the data may be highly multimodal. Hence, the generator needs to be very powerful/complicated to transform a unimodal gaussian into a distribution with multiple modes, as shown in Cornish et al 2020. One (unsatisfactory) solution is for the generator to just “collapse” to a small number of easy-to-produce modes, rather than covering all modes.
> GEBMs have an advantage over GANs in producing highly multimodal outputs. GEBMs allow rich posterior latent distributions that can be multimodal *in latent space* (by incorporating the energy function into the overall model). As a result, the generator doesn’t need to distort the inputs as much in order to produce multiple modes *in the observed space*. See the answer to reviewer 1 for more details.
> ### 2- Motivation of the density estimation experiment.
> Thank you, we will add the following clarification: We are only interested in this setting as a sanity check for the proposed learning method. This particular setting allows exact computation of the likelihood. Outside of this setting, closed form expressions of the normalizing constant are not available for generic GEBMs. Note that this is not an issue since the proposed method doesn’t require a closed form expression for the normalizing constant. However, in this experiment only, we want to have access to closed form expressions of the likelihood as a baseline against which to compare other density estimation methods.
> Details for training the NVP using CD are provided in Appendix  G.2:   It is trained as if the normalizing constant wasn’t known by running Langevin iterations to get samples from the model.
>
> This experiment is simply a sanity check and is not central to our setting, we are happy to move it to the appendix if the reviewers think it is more appropriate (the experiments were in fact added in response to a reviewer request from a submission to an earlier conference). We emphasize again that the main strength and intended purpose of the model is when the base measure is lower-dimensional than the ambient space.
>
>
> ### 3- “Experimental results are chaotic”. We respectfully disagree.
> We believe the experimental section 6 is structured and separated in two parts: Image generation and density estimation. Each part starts by describing the experimental setting then presents the result.
> We believe that we provided details about the architectures used for both the Generator and the Energy. We provide them again for the purpose of this discussion in the “Experimental details” response above.
>
> We are happy to provide any additional details upon request. We will also release the code for reproducing the experiments
>
> ### 4- Details of Figure 1 in section 2:
> We thank the reviewer for requesting the details of this illustrative example, which we think is of independent interest. We will provide those details in the appendix - for now we provide them in the “Experimental details” response above.

---

> > ### Comment · AnonReviewer2 · 2020-11-18
> > **Thanks!**
> >
> > Hi, all.
> >
> > Thanks for your kind response, which partially addresses my concerns about the experimental results and the toy examples.
> >
> > I still have the following concerns:
> >
> > 1. The mode collapse problem. Indeed, GEBM can model the distribution when the base distribution can "cover" the data distribution, i.e., the support of data distribution is a subset of the support of the base distribution. But when the mode collapse problem happens, i.e., the support of the base distribution is a subset of the support of the data distribution, then GEBM is impossible to model the data distribution. My question is: Can GEBM avoid this problem? Is the KALE objective function helpful to ensure that the support of the base distribution is a superset of the data distribution?
> >
> > 2. Many recent works treat GANs as EBMs and draw samples using MCMC, especially [*1]. Is GEBM trained with KALE better than [*1]?
> >
> > 3. In the toy example, it seems that the GAN is unable to recover the data distribution. GEBM uses more parameters to outperform GAN. If the GAN model is also able to recover the data distribution, i.e., trainable h(z) to transform z_2 instead of a linear transformation, will GEBM still outperforms GAN? This question is highly connected to the question "Why GEM is better than GAN". Can you provide an example that both GAN and GEBM can recover the data distribution, but GEBM is better than GAN?
> >
> > Overall, my concerns can be summarized as:
> >
> > Under the condition that both GEBM and GAN can model the data distribution, does GEBM have advantages over GAN, such as smaller model complexity, faster converges rate or better local minimum? If true, can you provide a toy example to demonstrate that?
> >
> > [*1] Grathwohl, Will, et al. "Your classifier is secretly an energy based model and you should treat it like one." arXiv preprint arXiv:1912.03263 (2019).

---

> > > ### Author Response · Authors · 2020-11-20
> > > **mode collapse and illustrative model 1/2**
> > >
> > > Thank you for the questions and the interesting discussion (and for the reply further up the page!)
> > >
> > > We'll address in order the two main points from your questions: the mode collapse, and the Figure 1 low dimensional  GEBM illustration.
> > >
> > > ## Mode collapse
> > > First, regarding the "mode collapse," we are completely in agreement: as you say, once the *generator* has chosen the support, and *if* the support of the generator is less than of the data, then the energy function *cannot* recover the lost support.  We would certainly not want to claim otherwise, since this would be completely wrong. We have emphasized this fact in revised text in the last paragraph of Section 2.
> > >
> > > What the energy function from the GEBM *does* achieve, given the support placed by the generator, is to refine the mass on this support.   This can be understood as follows: as long as the energy function isn't completely flat, it contains information about the residual mismatch in probability mass between the generator and target distribution, even after the generator has been trained to convergence. Thus, when the energy function isn't flat, it will improve the sample quality.  We have strong empirical evidence across multiple datasets that the energy function consistently improves sample quality over the generator alone. This is observed in Table 2, across CIFAR10, LSUN, Imagenet, and CelebA.  In Table 1, we see that the advantage is greater when the generator is simpler (SNGAN ConvNet vs ResNet). This finding is also consistent with the newly added result of proposition 3, which shows that the GEBM is closer to data than the generator of the GAN is.  As the GAN gets closer to the data, this gap reduces.
> > >
> > > Re your question "Is the KALE objective function helpful to ensure that the support of the base distribution is a superset of the data distribution?": this is an interesting question, however we would not want to claim this as an advantage as of now. It's entirely true that the KL is (informally speaking) "zero avoiding", so using the KL will tend to favor models whose support is a superset of the target. However the KALE is a variational lower bound on the KL, and it might not inherit this behaviour.
> > >
> > > In summary: We do *not* claim to avoid mode collapse with GEBMs. Rather, given the support placed by the generator, we *do* claim to better place mass on this support, compared with using the generator on its own.

---

> > > > ### Author Response · Authors · 2020-11-20
> > > > **mode collapse and illustrative model 2/2**
> > > >
> > > > ## Updates to toy model
> > > >
> > > > You are right that our current Figure 1 toy model misses an important point in a GEBM vs GAN comparison, which is that the generator had too few parameters to match the target. For this reason, we now propose an update to this toy illustration that better captures the GAN vs GEBM tradeoff, and answers your question: "If the GAN model is also able to recover the data distribution, i.e., trainable h(z) to transform z_2 instead of a linear transformation, will GEBM still outperforms GAN?"
> > > >
> > > > In our new experiment, we allow the generator to recover the data distribution by choosing the function h(z) to be a one hidden layer network instead of a simple linear transform. This network has universal approximation capability that depends on the number of units. This provides a direct control over the complexity of the generator. We varied the number of hidden units from 1 to 10^4 units.
> > > >
> > > > To evaluate generator performance vs generator complexity, we compared the performance of the GAN and GEBM using the Sinkhorn divergence (which estimates the Optimal transport distance) between each model and the data distribution. In all the experiments, we used the same discriminator/energy network. The results of these new experiments are provided in Figure 8 of the revised appendix G.4.
> > > >
> > > > From the experiments, we make the following observations:
> > > > 1- The GAN generator indeed improves when we increase the number of hidden units.
> > > > 2- The performance of the GEBM remains stable as the number of hidden units increases.
> > > > 3- The performance of the GEBM is always better than the GAN, although we can see the GAN converging towards the GEBM.
> > > > 4- GEBM with a simpler base already outperforms the GAN with more powerful generators.
> > > > 5- The gap between the GEBM and the GAN reduces as the GAN becomes more expressive.
> > > >
> > > > In summary: if you're training a GAN, then you have to learn an energy function for your critic, regardless of whether or not you choose to take advantage of it as in a GEBM. When the GAN generator is simpler, the GEBM energy function can provide a large boost to performance over the generator alone. As the generator becomes more complex, and gains the capacity to perfectly fit the target, then the benefit provided by the energy function decreases in the GEBM (the energy function will become flat, and will contain less useful information, as shown in Figure 8 of appendix G.4). From our results in Tables 1 and 2,  we are in the first scenario in practice, since we observe an advantage when we include the energy function.
> > > >
> > > >
> > > > ## Discussion of [1]:
> > > > From our understanding the model proposed in [1] does not use GANs. Instead the generative model is an EBM defined over the joint space of data and labels. Thus, it defines a density over the whole data space. One big strength of that model is that it allows to learn both a good EBM for the images and a good classifier for the label. Training is indeed performed using MCMC as in Du&Mordatch since it is an EBM with density defined over the whole space.
> > > > We will report the results from Table 1 in  [1]  where they obtained an FID score of 38.4 on Cifar10 which they compared to SNGAN (FID: 25.5).  Our results improve over these results.
> > > >
> > > >
> > > > We are happy to discuss further during the open period - please let us know if you have any further questions or comments, and we'll do our best to answer.

---

> > > > > ### Comment · AnonReviewer2 · 2020-11-21
> > > > > **Thanks**
> > > > >
> > > > > Thanks for your response. I'm happy to know that with such a large number of hidden units, GEBM is still better than GAN, with a large margin. I will raise my score to 6. However, I wonder if you use a deeper G instead of a wider G, will GEBM still outperform GAN? I think a deeper network can better utilize the parameters than a wider network.
> > > > >
> > > > > However, this paper is still not perfect. I think the following issues should be will-discussed in the final version:
> > > > >
> > > > > 1. Whether your KALE can help the mode collapse problem. If it is true, it will be a very strong advantage of your method: you can ensure that the base distribution can cover the support of the data distribution and the EBM can further refine it to achieve better performance. There are some papers that discuss the KL divergence and mode collapse problem in GANs like [*1] and [*2]. Maybe they are helpful.
> > > > >
> > > > > 2. The mechanism of why GEBM is better. Is it because using an EBM to refine the base distribution has better parameter efficiency? If not, why?
> > > > >
> > > > > 3. The computation efficiency of MCMC, or simply reject sampling.
> > > > >
> > > > > [*1] Dual Discriminator Generative Adversarial Nets. Tu Dinh Nguyen, Trung Le, Hung Vu, Dinh Phung
> > > > > [*2] Learning Implicit Generative Models By Teaching Density Estimators. Kun Xu, Chao Du, Chongxuan Li, Jun Zhu, and Bo Zhang

---

> > > > > > ### Author Response · Authors · 2020-11-24
> > > > > > **Thank you!**
> > > > > >
> > > > > > Thank you so much for the updated score and for your kind and encouraging response.
> > > > > >
> > > > > > To investigate the improvement due to using deeper generators,  we conducted a similar experiment as the previous one with a three hidden layer network each having N hidden units per layer.
> > > > > > We then varied the number of units from 10 to 5000 per layer and compared the performance. Due to the NxN matrices in the intermediate layers of the deep network, this corresponds to varying the number of parameters of the deep network from an order of magnitude of $10^2$ to $10^7$.
> > > > > > We included the result in Figure 8 of appendix G.4.  from which the same conclusions hold as before with the gap being further reduced as the number of parameters increases.
> > > > > >
> > > > > > 1- Alleviating Mode collapse using KL:
> > > > > > Thank you for the references, we added  the following discussion in the related work section:
> > > > > > “While we do not address the mode collapse problem, [1,2] showed that KL-based losses are resilient to mode collapse thanks to the zero-avoiding property of the KL, a good sign for KALE which is also derived from KL by Fenchel duality.“
> > > > > >
> > > > > > 2- Mechanism for why GEBM is better: We now clarified this in the last paragraph of section 2 and also added the following discussion in appendix G.4:
> > > > > > As long as the energy is not flat, it witnesses a remaining difference between the generator and training samples. This information allows the GEBM to do better than a GAN that ignores it. The performance gap between the GEBM and the GAN reduces as the generator becomes more powerful and forces the energy to be more flat as shown in Proposition 3.  This is also consistent with the results of Figure 8 and the experiments in section 6.
> > > > > >
> > > > > > 3- MCMC vs rejection sampling:
> > > > > > We now clarified this in section 4:  despite their algorithmic simplicity, methods such as rejection sampling are highly inefficient in high dimensions since a large number of proposed samples are likely to be rejected.  This low rejection rate has a direct impact on computational efficiency. MCMC methods were developed precisely to overcome those issues as discussed in [3] (section 1.3 (The sampling problem)) and  [4,5].
> > > > > >
> > > > > > Thank you again for your questions and insightful discussion!
> > > > > >
> > > > > >
> > > > > > References:
> > > > > >
> > > > > > [1] Dual Discriminator Generative Adversarial Nets. Tu Dinh Nguyen, Trung Le, Hung Vu, Dinh Phung
> > > > > >
> > > > > > [2] Learning Implicit Generative Models By Teaching Density Estimators. Kun Xu, Chao Du, Chongxuan Li, Jun Zhu, and Bo Zhang
> > > > > >
> > > > > > [3]:  M. Haugh. Mcmc and bayesian modeling. IEOR E4703 Monte-Carlo Simulation, Columbia University, 2017.
> > > > > >
> > > > > > [4] D. J. C. Mackay. Introduction to monte carlo methods. In Learning in graphical models, pages 175–204. Springer, 1998.
> > > > > >
> > > > > > [5] R. Y. Rubinstein and D. P. Kroese. Simulation and the Monte Carlo method, volume 10. John Wiley & Sons, 2016.

---

### Official Review · AnonReviewer4 · 2020-10-27
**reviewer 4**

**Rating:** 5
**Confidence:** 3

**Review:**

Summary: In this work, a generalized energy-based model (GEBM) is proposed. During the generation, the base distribution and the energy cooperate to combine the strengths of both the energy-based model and the implicit generative model.

+ves:

1. This paper has proposed a framework so that the energy function can be used to refine the probability mass on the learned base distribution. The framework is trained by alternating between learning the energy and the base. Empirically, the framework outperforms GAN with the same complexity.

2. KL Approximate Lower-bound Estimate (KALE) is used for energy training. There is a lot of detail on this derivation.

Concerns:
1. In this work, a new framework is proposed for training. The whole process seems complicated. Is there is a simple way to refine the probability mass on the learned base distribution? How about  considering exp(-E(x))G(x). Here G(x) is learned base distribution. A simple way to refine the probability mass is by sampling several generated output near the point and weight the outputs with exp(-E(x))G(x)? This seems easier. Could you explain the more possible benefits of your method?

2. During the generation process, two algorithms are proposed: ULA and KLA.  It seems a large variance. It is unclear which one is better.



Questions during the rebuttal period:
Please address and clarify the cons above:
Missed related work.

For the training of energy based models, there are several related work [2][3], etc.
In the work [1], the residual energy is used to better generation. MCMC is also used for generations.

[1] Residual Energy-Based Models for Text Generation. ICLR 2020

[2] Structured Prediction Energy Networks, ICML 2016

[3] Learning Approximate Inference Networks for Structured Prediction. ICLR 2018

---

> ### Author Response · Authors · 2020-11-13
> **Thank you for your review**
>
> Thank you for the insightful and constructive remarks and for pointing to the related works which we will include in the revised version. We hope the following discussion answers your questions.
>
> ### 1- What are the benefits of the proposed sampling compared to easier sampling methods?
> Thank you for pointing out this essential question.  In addition to the current discussion in the introduction and section 4, we will add a discussion in section 4 on the advantage of MCMC over importance sampling (IS), which is the alternative you suggested (if we understood correctly):
> Indeed, IS is a simpler way to estimate expectations under the GEBM by  sampling multiple points from the generator and then weighting using the energy. However, IS can lead to highly unreliable estimates, a well known problem in the Sequential Monte Carlo (SMC) literature which employs IS extensively [1,2]. Instead we use ULA/KLA which do not have these issues.
> The other advantages of the proposed method as discussed in section 4 and introduction are:
> - Latent sampling vs data-space sampling: Avoiding the curse of dimension.
> As shown in Prop 5,  sampling using methods that exploit the latent structure have a speed of convergence that depends only on the intrinsic dimension of the model (latent dimension) and not the ambient space dimension. This is a big advantage in practice for datasets like Images.
> - Using gradient information during MCMC (as done with ULA and KLA) is more efficient as it exploits the local slope of a density to find high density regions quickly.
>
> ### 2- Sampling Algorithms (ULA vs KLA) which one is better?
>
> This is an important question with implications even beyond the GEBM framework:
>  ULA is the Unadjusted langevin Algorithm which doesn’t use a momentum variable.
> KLA is a kinetic sampler, it uses a momentum variable just like Hamiltonian Monte Carlo. The strength of the momentum depends on a friction parameter $\gamma$. Higher friction leads to an algorithm that behaves like ULA. Lower friction leads to more exploration of the modes.
>
> - The experiments of Table 2, use KLA in a high friction regime thus behaving essentially as ULA. For this experiment,  we selected this high friction regime for KLA as we are interested in the sharpness of the generated images.
>
> - We also provide empirical evidence in figure 3, 4 and 5 of appendix D for the qualitative difference in behavior between both: We found that KLA (in a low friction regime) is able to explore multiple modes of the distribution in the same chain, unlike ULA which remains in the same mode. As discussed in section 4, this confirms prior theoretical and empirical results about KLA vs ULA.
>
> #### In conclusion:
> if the goal is to get sharper images and thus better FID scores, we recommend using either KLA with high friction or using ULA. If the goal is to explore multiple modes within the same MCMC chain, we recommend using KLA with a smaller friction parameter.
>
> #### References
> [1]: Del Moral, Pierre and Doucet, Arnaud and Jasra, Ajay, Sequential monte carlo samplers. Journal of the Royal Statistical Society: Series B (Statistical Methodology)
> [2]: Doucet, Arnaud and Freitas, Nando de and Gordon, Neil, Sequential Monte Carlo Methods in Practice

---

> > ### Author Response · Authors · 2020-11-20
> > **New illustrative experiment**
> >
> > We have added a new toy illustration on the difference between EBM and GEBM that demonstrates clearly where GEBM is better. please see the new response to reviewer 2 for details.
> >
> > We are happy to discuss further during the open period - please let us know if you have any further questions or comments, and we'll do our best to answer.

---

### Official Review · AnonReviewer1 · 2020-10-29
**Good paper, interesting but not suprising model, wish to see more comparisons with GAN**

**Rating:** 6
**Confidence:** 4

**Review:**

Originality:
The paper presents an interesting generative model termed generalized energy-based model (GEBM), which is essentially an exponential tilting of a plain generator model P_G by an energy based model exp(-E).
The appearance of such a model in generative modeling looks to be new. But the structure of the model is not surprisingly new.

Significance:

The pros:
1. The purpose of introducing such a model, as stated by the authors, is to make use of the fact that target distribution may concentrate in a low dimensional manifold in the target domain, which can be captured by a low-dimensional generator model P_G. Once tilted by exp(-E), the overall model can generate sharper samples than the plain energy based model.
2. The paper also presents tractable methods to train and sample from the proposed GEBM.
3. The authors show that GEBM performs better than GANs on certain image generation experiments.

The cons:

While it may be easier to understand why GEBM can be better than EBM on certain generative modeling tasks, the authors did not provide much explanation or justification on why GEBM can outperform GANs. As mentioned by the authors, the energy based models and functional generators are quite different generative models. By combining them, how to guarante that their pros rather than the cons will be strengthened? In what kind of tasks GEBM can perform better than GANs? Hopefully the authors can give more theoretical comparisons or intuitions to address these questions.


Quality:

Pros:
The overall quality of the paper is good. Nearly all results regarding the properties of GEBM come with theoretical justifications.
The proposed alternative training method and sampling methods look reasonable.

Cons:
It would be better if there is some analytical comparison between GEBM and GAN.
Also, in the experiment section, the authors should reveal more detail on how the energy function (class) is chosen.
Another concern is that, for Proposition 1 to hold, does it require the generator G to be invertible?

---

> ### Author Response · Authors · 2020-11-13
> **Thank you for your review**
>
> Thank you for the encouraging and constructive remarks and for the insightful questions.
> ### 1- Exponential tilting interpretation:
> We really like the interpretation of GEBMs as an exponential tilting of a reference measure ( here  the generator). We will add it to section 2, further emphasizing how GEBM extends exponential tilting models by allowing the reference measure ( the generator) to change its support /shape in space. This enables the GEBMs to learn the manifold of low dimensional data in addition to the tilting of mass on that manifold. With this new flexibility comes the question of learning reference measure as well, which is what we propose here.
> ### 2-  Theoretical advantage of GEBM relatively to GAN:
> This is an important question which encouraged us to strengthen our theory results. We will add a discussion on this matter in section 2 and add a proposition which provides insights on this question, which we summarize in 3 points:
> #### 1- The proposition:
> It considers the case when the generator is trained until it matches the support of the data. In this case, the KL between data $P$ and generator $G$ is defined and we show that $KL(P| Q) <=  KL(P|G)$, where $Q$ is the GEBM with the generator $G$ as a base. The best case scenario is when the exponential tilting is equal to the density ratio between P and G, in which case $KL(P| Q)=0$. This means that the exponential tilting of the generator $G$ systematically improves over using the generator alone.
> #### 2- When can we expect most improvement over GAN?
> According to the above result, if $G$ is exactly equal to the data distribution $P$, then no further improvement is possible. But as long as there is an error in mass on this common support, the GEBM improves over the generator $G$.
> It may be that, apart from the error in mass on the support, the support itself is not correctly obtained. Even in this case, the addition of an energy function can improve over the generator alone. This is consistent with Table 1 which shows that the improvement is larger for the smaller networks (SN-GAN ConvNet) than for the larger one  (SN-GAN ResNet). Thus, we expect the highest improvement when the generator has a limited strength.
> #### 3- Multimodality : GEBMs vs GANs
> In GANs, the latent noise is usually unimodal (a gaussian for instance), however the data is often highly multimodal. Hence, the generator needs to be very powerful/complicated to transform a simple gaussian into a distribution with multiple modes as shown in Cornish et al 2020. On the other hand, EBMs allow to capture multimodality with rather simpler models.  GEBMs build on this intuition and allow to use richer latent distributions (thanks to proposition 1) that can be multimodal in latent space. The energy removes the additional burden from the generator of learning how to put weights on the support, so that it only focuses on getting the support correct. For instance, Table 1 shows that a smaller generator network (SNGAN ConvNet) combined with an energy achieves a better FID score on imagenet (FID = 13.94) compared to a GAN using a larger generator SNGAN ResNet (FID=20.5). This improvement is observed on Imagenet which has many more classes than Cifar10. As such, we expect it to be much more multimodal. Thus we think that GEBMs are the most useful in high multimodality scenarios. (Please see also response to AnonReviewer2.)
>
> #### Summary:
> While these intuitions and empirical observations are favorable to GEBMs,  a precise theoretical quantification requires a number of steps to be taken:
> - Identifying the modeling capacity of GANs and the additional capacity that the energy provides.
> - Providing a Generalization theory for f-divergences which are harder to analyze than IMPs (Uppal 2019).
> - Extending the generalization theory to the case where density ratios are not well defined. We provided analysis when the ratio is well-defined in Theorem 6 of  appendix B.
>
> All those questions are challenging and interesting future research directions.
> ### 3- Network Architectures:
> Thank you, we will include details for the illustrative example in figure 1 of section 2  (please see the “Experimental details” response above for those details), we believe we provided details for the networks used in section 6, but we are happy to include any additional details. We will also release the code for reproducing the experiments.
> ### 4- Does the generator need to be invertible for proposition 1 to hold?
> It might seem surprising at first, but this proposition doesn’t require invertibility of the generator. Proposition 1 says that exponential tilting of the generator is equivalent to an exponential tilting in latent space before mapping to data space. To prove this result (in appendix C.1), we rely on a characterization of probabilities through test functions: two probabilities are equal if they have the same expectation for any bounded continuous function.

---

> > ### Author Response · Authors · 2020-11-20
> > **Multimodality: GEBMs vs GANs**
> >
> > Regarding this point, please see also the new response to AnonReviewer2  and the revised version which includes an additional experiment  in Figure 8 of appendix G.4 and discussion.
> > We are happy to discuss further during the open period - please let us know if you have any further questions or comments, and we'll do our best to answer.

---

### Author Response · Authors · 2020-11-13
**Experimental Details**

Here we summarize some experimental details of section 6 for the purpose of the discussion and provide additional experimental details for the illustrative example of Figure 1 in section 2.


## 1- Experimental details for the experiments in section 6:

### For image generation:
We used the network architectures from Miyato and al (2018) which come in two versions one with convnets and another with ResNets  (we called them SNGAN Convnet and SNGAN ResNet). For the base, we use their generators networks and for the energy we used their discriminator networks.
### For density estimation:
We used a real NVP network for both base and energy and we referred to Appendix 5.2 for the details of those networks which are: 5 NVP layers consisting of a shifting and  a scaling layer. Each of those shifting and scaling layers consist of two hidden layers of 100 neurons.

## 2- Details of Figure 1 in section 2:
We will provide those details in the appendix:
### Data:
Sample $Z$ uniformly from $[0,1]$. Apply a fixed invertible transform $X_1 = h(Z)$  to get a bimodal distribution. Generate data $(X_1,X_2)$ using a function $X_2 = G_{\theta^{\star}}(X_1)$.
- The transform $h(z)$ is a sigmoid function
- The parametric function $G$ is given by $G_{\theta}(x) = sin(Wx)/(1+(Bx)^{2})$

### GAN:
-Generator: Sample $Z$ uniformly from $[0,1]$. Generate  sample $(X_1,X_2) = ( G_{\theta}(Z),     WZ+b )$. Learn $\theta$
-Discriminator: MLP with 3 layers and hidden 10 units.

### GEBM:
Base is the same as the generator of the previous GAN. Energy is the same as the Discriminator network of the previous GAN.
### EBM:
energy of the form: $X_2$ conditionally on $X_1$ is a gaussian with mean $G_{\theta}(X_1)$ and trainable variance $\sigma^2$. $X_1$ is a mixture of gaussians. EBM is learned using Maximum likelihood since the normalizing constant is tractable.

Both GAN and GEBM have the capacity to recover the the exact support by finding the optimal parameter $\theta^{\star}$. EBM has the capacity to learn a conditional gaussian $X_2|X_1$  whose mean matches the support of the data.

---

### Public Comment · ~Jianwen_Xie1 · 2020-11-14
**missing related references**

Dear Authors and Reviewers,

We found that the current paper missed some important references about pioneering works that are related to energy-based generative models parameterized with deep net energy.

The first paper that proposes to train an energy-based model parameterized by modern deep neural network and learned it by Langevin based MLE is in (Xie. ICML 2016) [1]. The model is called generative ConvNet, because it can be derived from the discriminative ConvNet. This is also the first paper to formulate modern ConvNet-parametrized EBM as exponential tilting of a reference distribution, and connect it to discriminative ConvNet classifier. That is, EBM is a generative version of a discriminator.
(Xie. ICML 2016) [1] originally studied such an EBM model on image generation theoretically and practically in 2016.

(Xie. CVPR 2017) [2] (Xie. PAMI 2019) [3] proposed to use Spatial-Temporal ConvNet as the energy function in EBMs for video generation. The model is called Spatial-Temporal generative ConvNet.

(Xie. CVPR 2018) [4] also proposed to use volumetric 3D ConvNet as the energy function for 3D shape pattern generation. It is called 3D descriptor Net.

Also, the Generative Cooperative Nets (CoopNets) (Xie. PAMI 2018)[5] and (Xie. AAAI 2018) [6], which jointly trains an EBM and a generator network by MCMC teaching.

Those are the more original and earlier papers for deep EBMs with ConvNet as energy function than what you have cited, e.g., [7](Yilun Du and Igor Mordatch, 2019).


References:

[1] A Theory of Generative ConvNet. Jianwen Xie *, Yang Lu *, Song-Chun Zhu, Ying Nian Wu (ICML 2016)

[2] Synthesizing Dynamic Pattern by Spatial-Temporal Generative ConvNet Jianwen Xie, Song-Chun Zhu, Ying Nian Wu (CVPR 2017)

[3] Learning Energy-based Spatial-Temporal Generative ConvNet for Dynamic Patterns Jianwen Xie, Song-Chun Zhu, Ying Nian Wu IEEE Transactions on Pattern Analysis and Machine Intelligence (TPAMI) 2019

[4] Learning Descriptor Networks for 3D Shape Synthesis and Analysis Jianwen Xie *, Zilong Zheng *, Ruiqi Gao, Wenguan Wang, Song-Chun Zhu, Ying Nian Wu (CVPR) 2018

[5] Cooperative Training of Descriptor and Generator Networks. Jianwen Xie, Yang Lu, Ruiqi Gao, Song-Chun Zhu, Ying Nian Wu. IEEE Transactions on Pattern Analysis and Machine Intelligence (TPAMI) 2018

[6] Cooperative Learning of Energy-Based Model and Latent Variable Model via MCMC Teaching. Jianwen Xie, Yang Lu, Ruiqi Gao, Ying Nian Wu. AAAI 2018.

[7] Yilun Du and Igor Mordatch. Implicit generation and modeling with energy based models. In Advances in Neural Information Processing Systems, pages 3603–3613, 2019

Thank you!

---

> ### Author Response · Authors · 2020-11-16
> **Thank you!**
>
> Dear Jianwen Xie,
> Thank you for your interest in our work and for bringing these works to our attention!
> We have uploaded a revised version of the paper which now includes references to the papers you proposed, and discusses them as follows:
> - In section 2.1, we refer to [1] for the exponential tilting interpretation and then further emphasize that GEBM provides an additional flexibility in allowing the reference measure (the generator) to change its low dimensional shape.
> - In the related work section we include now [1,2,3,4] as prior works developing state-of-the-art EBMs and methods for training them.
> - In the same section, we discuss [5,6] as follows:
> "In [5,6], two independent models, a full support EBM and a generator network, are trained cooperatively using MCMC. By contrast, in the present work,  the energy and base are part of the same model, and the model support is lower-dimensional than the target space $\mathcal{X}$.   “
>
> Thank you!

---

> ### Comment · AnonReviewer2 · 2020-11-18
> **Not related**
>
> Were the papers you linked to focusing on defining density on the low dimensional manifold?
>
> Simply citing (Yilun Du and Igor Mordatch, 2019) is enough to provide a background for this paper.
>
> Most of the linked papers are not related to this paper.

---

### Author Response · Authors · 2020-11-16
**Revised version of the submission**

We thank the reviewers for their insightful comments and for encouraging us to strengthen the discussions and presentation of the paper. We now updated a revised version of the paper which includes the suggestions of the reviewers and clarifications/answers. We hope this new version addresses all reviewers concerns.  We summarize them as follows:

1.  Advantages of GEBM:
    - We included a section 2 to discuss the advantages of GEBM over GANs and EBM. This one is based on the responses to AnonReviewer1  AnonReviewer2  and contains:
        - The exponential tilting interpretation
        - The advantage over GAN in the setting of multimodal data
    - In section 3.1, we inlucuded a new proposition 3, based on the response to AnonReviewer1.
    - We clarified in section 2, that proposition 1 doesn't need the invertibility of the generator.
2.  Advantages of the sampler:
    - We included a discussion at the beggining of section 4. to discuss the advantage of the MCMC latent sampling over Importance sampling, based on the response to AnonReviewer4.
    - We included a discussion of the differences between ULA and KLA at the end of the experiments section 6.1 based on the response to AnonReviewer4.
3.  Experiments section 6:
    - Motivation of the density estimation task:
        - We included a paragraph in section 6.2 to clarify the motivations of this particular experimental setting.
4.  Experimental details:
    - We included the details of the illustrative figure in Appendix G.3
    - We included a full description of the network architectures used for the image generation task in Appendix G.1 (Tables 5 to 8).
5.  References:
    - We included the references suggested by AnonReviewer4 in the related work section.
    - We included  the references suggested in the comment of Jianwen Xie in the related work section.

---

### Author Response · Authors · 2020-11-20
**Revised version of the submission**

We have updated the submission to include an additional experiment in response to AnonReviewer2. This experiment builds on the illustrative example of figure 1 and allows the generator to recover the data distribution using a richer model. We then evaluate the generator performance vs generator complexity (as measured by the number of hidden units) and compare with GEBMs. The results of this experiment are shown in Figure 8 of appendix G.4 and discussed there.

We also updated the last paragraph of section 2 to refer to those results and discuss them.
Please see also the new response to AnonReviewer2.

We are happy to discuss further during the open period - please let us know if you have any further questions or comments, and we'll do our best to answer.

Thank you!

---

### Author Response · Authors · 2020-11-24
**Revised version of the document**

We have updated the submission to include an additional experiment in response to AnonReviewer2. This experiment further investigate the improvement due to using deeper generators instead of shallow. It uses a similar setting as the previous one and also builds on the illustrative example of figure 1. We then evaluate the generator performance vs generator complexity (as measured by the number of parameters) and compare with GEBMs. The results of this experiment are shown in Figure 8 of appendix G.4 and discussed there.

Thank you!

---

### Decision · Program_Chairs · 2021-01-07
**Final Decision**

**Decision:**

Accept (Poster)

**Comment:**

This paper introduces a generative model termed generalized energy-based model (GEBM).

The goal is modelling complex distributions supported on low-dimensional manifolds, while offering more flexibility in refining the distribution of mass on those manifolds. The key idea is presented as parametrizing the base measure (called a generator in the paper) and the density with respect to this base measure separately. Figure 1 of the paper sketches the idea on a very clear toy example.

The pros:
* Flexibility: Decomposing the full problem as learning the support and learning the density on this support
* Theoretical justification
* Introducing the KALE objective
* Comparative empirical results with GANs show the additional benefits. Empirically, the framework outperforms GAN with the same complexity.
* Clear written paper

The lack of a comparison with GANs has been raised as a concern.  The authors have satisfactorily answered key questions and
others raised during rebuttal and added several new references. They have also improved the narrative and included an additional experiment to contrast GEBM and GANs in response to AnonReviewer2, also provided more detail
on how the energy function (class) is chosen.